# Temporal coordination of the transcription factor response to H₂O₂ stress

Elizabeth Jose [1], Woody March-Steinman [2], Bryce A. Wilson[1], Lisa Shanks[1], Chance Parkinson [1], Isabel Alvarado-Cruz[3], Joann B. Sweasy[3,4,5] & Andrew L. Paek [1,2,4] ✉

Oxidative stress from excess H₂O₂ activates transcription factors that restore redox balance and repair oxidative damage. Although many transcription factors are activated by H₂O₂, it is unclear whether they are activated at the same H₂O₂ concentration, or time. Dose-dependent activation is likely as oxidative stress is not a singular state and exhibits dose-dependent outcomes including cell-cycle arrest and cell death. Here, we show that transcription factor activation is both dose-dependent and coordinated over time. Low levels of H₂O₂ activate p53, NRF2 and JUN. Yet under high H₂O₂, these transcription factors are repressed, and FOXO1, NF-κB, and NFAT1 are activated. Time-lapse imaging revealed that the order in which these two groups of transcription factors are activated depends on whether H₂O₂ is administered acutely by bolus addition, or continuously through the glucose oxidase enzyme. Finally, we provide evidence that 2-Cys peroxiredoxins control which group of transcription factors are activated.

Hydrogen peroxide (H₂O₂) is a reactive oxygen species (ROS) with a complex role in cellular physiology. H₂O₂ is produced as a byproduct of cellular respiration and by over 40 enzymes in humans[1]. H₂O₂ functions as a second messenger, activating signaling pathways that promote proliferation, differentiation, and wound healing[2-5]. Yet at high concentrations, H₂O₂ is toxic to cells due to the creation of hydroxyl radicals by the Fenton reaction. Hydroxyl radicals cause DNA damage, lipid peroxidation, and the formation of unfolded/aggregated proteins, all of which inhibit cell proliferation and can induce cell death[6,7]. Thus, H₂O₂ levels must be tightly regulated and rapidly cleared when concentrations are too high, and elevated levels of H₂O₂ are thought to be the underlying cause of many pathologies[8].

To counter high levels of H₂O₂, metazoans activate several transcription factors (TFs) including p53, FOXO, NRF2 and other TFs, which act to restore the redox state of the cell and repair damage caused by oxidative stress[9]. Upon activation by H₂O₂, these TFs upregulate hundreds of target genes in diverse cytoprotective processes including cell-cycle arrest, NADPH/GSH production, ROS scavenger enzymes, DNA damage repair, autophagy, and protein quality control[10-15]. In addition to their cytoprotective role, both p53 and FOXO can induce cell death by upregulating apoptotic genes[16,17].

Given the diverse molecular challenges caused by oxidative stress, which TFs are activated, and their order of activation is likely tightly regulated and dependent on H₂O₂ concentration[18]. Indeed, oxidative stress is often differentiated broadly into either eustress (mild oxidative stress) or distress (toxic oxidative stress) and it is thought that these different levels of stress activate distinct TFs[1,19]. Yet which TFs are activated at low vs. high oxidative stress and the relative timing of TF activation is not known and is essential for understanding how cells combat oxidative stress and how this breaks down in disease.

In yeast, the timing of TF activation is tightly controlled and dependent on H₂O₂ concentration. This is best illustrated by Pap1, a TF activated by H₂O₂ stress in fission yeast[20,21]. Pap1 is activated by Tpx1, a 2-Cys peroxiredoxin (PRDX) protein. Tpx1 activates Pap1 through a redox-relay mechanism, where oxidative equivalents stemming from H₂O₂ are passed from Tpx1 to a cysteine in Pap1[20].

[1]Molecular and Cellular Biology, University of Arizona, Tucson, AZ 85721, USA. [2]Program in Applied Mathematics, University of Arizona, Tucson, AZ 85721, USA. [3]Cellular and Molecular Medicine, University of Arizona College of Medicine, Tucson, AZ 85724, USA. [4]University of Arizona Cancer Center, Tucson, AZ 85724, USA. [5]Fred and Pamela Buffett Cancer Center and Eppley Institute for Cancer Research, University of Nebraska Medical Center, Omaha, NE 68198, USA. ✉e-mail: apaek@arizona.edu

This leads to the formation of an intramolecular disulfide bond in Pap1, and Tpx1 further promotes this bond by oxidizing thioredoxin[22]. The Pap1 disulfide bond causes nuclear accumulation of Pap1 and activation of downstream target genes. Pap1 nuclear accumulation occurs rapidly at low levels of $H_2O_2$, yet at higher concentrations there is a delay in Pap1 activation[23,24]. The delay at high $H_2O_2$ is due to hyperoxidation of a key cysteine residue in Tpx1, which blocks Tpx1 dependent redox relays. Hyperoxidation of Tpx1 can be reversed by the sulfiredoxin enzyme (Srx1 in yeast, SRXN1 in humans), but this takes time, and thus Pap1 activation is delayed until Srx1 repairs hyperoxidized Tpx1[25].

Little is known about the temporal regulation of $H_2O_2$ induced TF activation in mammals. Yet the conservation of key redox regulatory proteins, including the PRDX/SRXN1 system, suggests temporal regulation of TFs in response to $H_2O_2$ is likely. Similar to yeast, there is strong evidence that PRDX dependent redox relays occur in mammals. Gene knockout models of *PRDX1* and *PRDX2* in HEK293T cells showed reduced protein disulfide bond formation following oxidative stress, and transient disulfide bond intermediates have been recovered between 2-Cys PRDX proteins and hundreds of other proteins, suggesting that PRDX-dependent redox relays control the oxidation state of a large body of proteins[26,27]. Inactivation of PRDX proteins by hyperoxidation also occurs in response to high $H_2O_2$ levels, suggesting $H_2O_2$ concentration dependent signaling. Further evidence supports a role for PRDX proteins in regulating TFs. For example, PRDX2 regulates STAT3 by a redox relay resulting in a disulfide bond in STAT3 causing oligomer formation and attenuation of transcription[28,29]. PRDX1 can form disulfide bonds with FOXO3, which leads to its retention in the cytosol[30,31]. Yet the role of PRDX proteins in regulating the timing of TF activation in response to $H_2O_2$ and how the timing of activation is affected by dose is unclear.

In this study we find that the specific TFs activated by $H_2O_2$, and their activation timing, depend on the $H_2O_2$ concentration and method of $H_2O_2$ delivery (acute vs continuous). We first focus on FOXO1 and p53 as both are activated by $H_2O_2$ and upregulate genes in many overlapping pathways including cell-cycle arrest and apoptosis[16,17,32,33]. Using immunofluorescence and time-lapse imaging we find that low levels of bolus $H_2O_2$ treatment cause an immediate increase in p53 levels while FOXO1 remains inactive in the cytoplasm. At higher bolus $H_2O_2$ concentrations there are two temporal phases of activation: in the first phase FOXO1 is shuttled to the nucleus within one hour, while p53 levels are kept low. In the second phase, FOXO1 exits the nucleus which is followed by an increase in p53 levels. The duration of the first phase, where FOXO1 is active and p53 inactive, increases with $H_2O_2$ dose. Interestingly, if $H_2O_2$ is produced continuously by the enzyme glucose oxidase, the order of activation is reversed with p53 accumulation preceding nuclear FOXO1 shuttling. Focusing on bolus treatment, we find that other TFs are activated either with FOXO1 (NF-κB, NFAT1) or with p53 (NRF2, JUN) but not both, suggesting coordinated regulation of each group of TFs. The difference in TF activation between the two temporal phases is reflected in large differences in gene expression with increases in ribosome, oxidative phosphorylation, and proteasome genes in phase 1, and NRF2 and p53 target genes involved in NADPH, glutathione, and nucleotide production increasing in phase 2. Finally, we provide evidence that the peroxiredoxin/sulfiredoxin system controls which group of TFs is activated. The distinct target genes activated in each phase, coupled with the evolutionary conservation of a PRDX control mechanism, suggests that activating specific TFs at distinct concentrations of $H_2O_2$ is critical for properly restoring redox balance.

## Results

### Mutually exclusive activation of FOXO1 and p53 in response to acute $H_2O_2$ stress

To determine if FOXO1 and p53 are activated at the same level of $H_2O_2$ stress, we performed a dose response in MCF7 cells and measured FOXO1 and p53 five hours after treatment by immunofluorescence. FOXO1 is regulated by nuclear/cytoplasmic shuttling, so we measured FOXO1 activation by determining the fraction of nuclear FOXO1 in individual cells[34]. For p53 we measured mean nuclear levels as p53 is predominantly localized to the nucleus and increases due to inhibition of MDM2 dependent proteasomal degradation[35]. At low $H_2O_2$ concentrations nuclear FOXO1 levels were unchanged yet p53 levels increased (Fig. 1a–c, 20–60 μM). At higher $H_2O_2$ concentrations we observed two distinct populations of cells: one population with increased p53 levels and cytoplasmic FOXO1, and a second population with predominantly active (nuclear) FOXO1 and low p53 levels (Fig. 1a–c, 80–100 μM, see Supplementary Fig. 1A for activation thresholds). The proportion of cells with nuclear FOXO1 increased with $H_2O_2$ concentration, while cells with high p53 levels decreased. At the highest $H_2O_2$ dose (Fig. 1a–c, 200 μM), most cells had active FOXO1, while p53 active cells were comparable to untreated controls. Cells with activation of both FOXO1 and p53 were <5% in all doses tested, suggesting that activation of FOXO1 and p53 is mutually exclusive in response to acute $H_2O_2$ stress.

The response to $H_2O_2$ is known to depend on the number of cells in an experiment[36], and indeed we observed the concentration of $H_2O_2$ required to activate FOXO1 increased with the number of cells plated (Supplementary Fig. 1B). Therefore, for all experiments in this study we took care to plate equal numbers of cells in control and treatment groups to properly measure relative activation of each TF.

The lack of p53 accumulation at high doses of $H_2O_2$ was unexpected as $H_2O_2$ causes DNA damage, which in principle should activate p53[37]. To ensure that $H_2O_2$ was inducing DNA damage, we repeated the $H_2O_2$ dose-response and measured p53 levels and phosphorylation of serine 139 of histone H2AX (γH2AX), a marker for DNA-damage, in single cells using immunofluorescence. As expected, $H_2O_2$ induced a dose-dependent increase in γH2AX levels (Fig. 1d) and the formation of γH2AX foci (Supplementary Fig. 1D), with the highest dose (200 μM) causing the largest increase in γH2AX. In contrast, p53 levels showed the highest increase at intermediate doses of $H_2O_2$ (50–100 μM) but were comparable to untreated controls at doses >= 150 μM (Fig. 1e). Levels of γH2AX at higher doses of $H_2O_2$ were comparable to treatment with the DNA damaging agent Neocarzinostatin (NCS) (Supplementary Fig. 1C). Increased γH2AX can occur in the absence of DNA damage, thus, to ensure that $H_2O_2$ was inducing DNA damage we used a comet assay[38]. We observed a dose dependent increase in the percentage of DNA in the tail for both the alkaline comet assay (Fig. 1f), which measures both single and double strand breaks, and in the neutral comet assay (Fig. 1g) which measures double strand breaks. Finally, high $H_2O_2$ concentrations suppressed p53 accumulation by NCS (Fig. 1h). The lack of p53 accumulation despite high levels of DNA damage suggests a control mechanism to actively block p53 accumulation at high concentrations of $H_2O_2$.

We next measured how protein and lipid oxidation scales with the concentrations of $H_2O_2$ tested. For protein oxidation, we measured maleimide-488 incorporation in proteins and normalized to total protein content[39]. Maleimide-488 incorporation decreased at all doses tested, indicating a decrease in reactive thiol groups and high levels of protein oxidation (Supplementary Fig. 1M). In contrast, lipid peroxidation, as measured by 4-HNE staining increased only at high concentrations of $H_2O_2$ (150 μM or more, Supplementary Fig. 1N).

Mutually exclusive activation of FOXO1 and p53 in response to $H_2O_2$ is not limited to MCF7 cells as we observed the same pattern in MCF10A, A549 and U2OS cell lines (Supplementary Fig. 1E–J). However, the U2OS and A549 cell lines only activated FOXO1 at high levels of $H_2O_2$ (500 μM). A549 cells harbor mutations in KEAP1, which block its interaction with NRF2, leading to constitutive activation of NRF2 and is likely the reason this cell line is more resistant to $H_2O_2$[40]. It is not clear why U2OS cells are more resistant to $H_2O_2$. The oxidative stress inducing agent menadione also induces mutually exclusive activation of FOXO1 and p53 (Supplementary Fig. 1K). However,

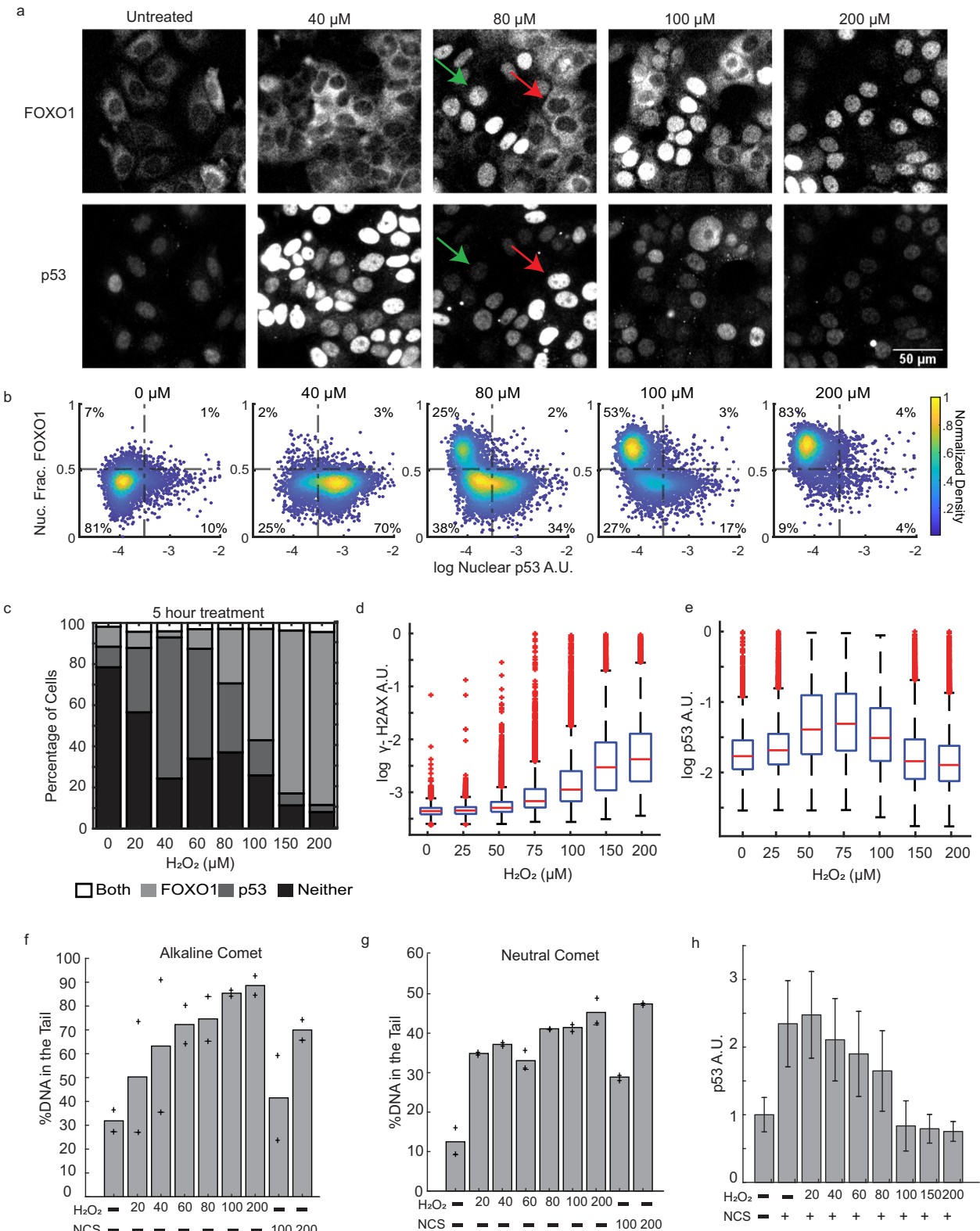

tert-butyl hydroperoxide activated p53 but not FOXO1 suggesting a different mode of activation (Supplementary Fig. 1L). Menadione induces ROS through redox cycling which creates superoxide radicals that dismutate to $H_2O_2$, and menadione induced cell death can be suppressed by overexpressing catalase[41,42]. Thus, it is possible that mutually exclusive activation of p53 and FOXO1 is specific to $H_2O_2$. Together these data suggest that mutually exclusive activation of p53

and FOXO1 is not cell-type specific but does not occur under all forms of oxidative stress.

## FOXO1 activation precedes p53 activation at high concentrations of acute $H_2O_2$ stress

The surprising lack of p53 activation at high $H_2O_2$ concentrations might be due to a temporal delay in activation as our

**Fig. 1 | Mutually exclusive activation of FOXO1 and p53 in response to H₂O₂.**
**a** Immunofluorescence images of MCF7 cells treated with indicated concentrations of H₂O₂ for 5 h and stained for FOXO1 (top row) and p53 (bottom row). Green arrow indicates a cell with nuclear FOXO1 and low levels of p53. Red arrow shows a cell with cytoplasmic FOXO1 and increased levels of p53. Experiments were repeated over 10 times with similar results. **b** Density colored scatter plots ($n \geq 2000$ cells) of the log of nuclear p53 levels (x-axis) and the nuclear fraction of FOXO1 (y-axis). **c** Percentage of cells activating both FOXO1 and p53, only FOXO1, only p53 and neither at the indicated concentrations at 5 h of H₂O₂ treatment. Thresholds are indicated by dashed lines in (**b**). **d** Box and whisker plots of γH2AX and (**e**) nuclear p53 levels measured by immunofluorescence ($n \geq 10,000$ cells) after 3 h of H₂O₂

treatment at indicated concentrations. γH2AX and nuclear p53 levels were measured in the same experiment. The central line indicates the median, bottom and top edges of the box are the 25th and 75th percentiles respectively. The whiskers indicate the extreme data points not considered as outliers and outliers are indicated by a red plus sign. **f** Median tail DNA content for alkaline and (**g**) neutral comet assay for indicated H₂O₂ concentrations (μM) and NCS concentrations (ng/mL). Crosses are values for 2 replicates. **h** Median p53 intensities measured by immunofluorescence for cells ($n \geq 3000$ cells) treated with NCS (+ = 800 ng/mL) and indicated concentrations of H₂O₂ (μM). Error bars are the median absolute deviation. A.U. Arbitrary Units. Nuc. Frac. - Nuclear Fraction. Source data are provided in Source Data Fig. 1.

immunofluorescence experiments were performed five hours after H₂O₂ treatment. To test this hypothesis, we tagged FOXO1 and p53 genes with fluorescent reporters to measure activation of both TFs in single cells over time. We used a previously developed MCF7 cell line where CRISPR/Cas9 was used to tag the endogenous locus of FOXO1 at the C-terminus with the mVenus fluorescent protein. This cell line also contains a H2B-ECFP tag for tracking nuclei[43]. For p53, we added an exogenously expressed p53-mCherry reporter used and validated in previous studies[44]. We then treated these cells with bolus treatment of four different concentrations of H₂O₂ and measured TF levels in single cells every 20 min for 24 h. At the lowest H₂O₂ dose (50 μM), FOXO1 remained largely inactive (in the cytoplasm), while p53 levels oscillated, as shown previously in response to DNA double strand breaks and H₂O₂ (Fig. 2a, b, Supplementary Movie 1)[44,45].

At higher H₂O₂ concentrations, FOXO1 accumulated in the nucleus within 1 h of treatment in a subset of cells (Fig. 2a, c–e, Supplementary Movies 2, 3, 4). While FOXO1 was in the nucleus, p53 levels remained low, and only increased after FOXO1 exited the nucleus. The fraction of cells with nuclear FOXO1 and the time in which FOXO1 remained in the nucleus increased with H₂O₂ dose. To visualize data from multiple cells, we created single-cell heat maps of nuclear FOXO1 and p53, where cells were sorted by the time in which FOXO1 remained in the nucleus (Fig. 2b–e). Aligning trajectories to the time that FOXO1 exited the nucleus revealed that p53 began accumulating ~1 h after FOXO1 exited the nucleus (Fig. 2f). Together these data revealed that there are two temporal phases to the FOXO1/p53 response to high concentrations of acute H₂O₂ stress. In the initial phase (phase 1), FOXO1 enters the nucleus and p53 levels remain low. In the second phase (phase 2), FOXO1 exits the nucleus and p53 begins to accumulate within 1 h.

The dynamics of p53 accumulation also differ in response to higher concentrations of H₂O₂. In some cells p53 levels oscillate similar to the 50 μM dose, yet often with a higher initial spike in p53 levels (Fig. 2a). While other cells show large bursts of p53 levels similar to the response to UV irradiation[46]. The proportion of cells with oscillating p53 levels decreased with dose as shown by autocorrelation analysis, and recently observed in retinal pigment epithelial cells (Supplementary Fig. 2a–d)[44]. These differences correlated with cell survival as the p53 levels in cells that died reached higher levels than those that survived (Supplementary Fig. 2G, H). In addition, autocorrelation analysis revealed oscillations in p53 activation in surviving cells but not in dying cells (Fig. 2g). Dying cells also showed an increase in the duration of FOXO1 activation as compared to surviving cells, in agreement with our previous study (Supplementary Fig. 2E, F)[43]. Together these data show that higher concentrations of H₂O₂ cause prolonged activation of FOXO1, a delayed yet stronger p53 response, and an increase in cell death.

To test the generality of the two temporal phases of FOXO1 and p53 activation, we used MCF10A cells, a non-cancerous breast epithelial cell line. We used CRISPR/Cas9 gene editing to tag FOXO1 with mVenus fluorescent protein, and incorporated p53-mCherry and H2B-ECFP fluorescent reporters. Similar to MCF7 cells, few cells exhibited

nuclear FOXO1 accumulation at 50 μM H₂O₂ (Supplementary Fig. 2I–K), and the fraction of cells with nuclear FOXO1 and the time it remained in the nucleus increased with H₂O₂ dose (Supplementary Fig. 2L–Q). Moreover, p53 accumulation was delayed at higher doses until after FOXO1 exited the nucleus.

## p53 accumulation precedes FOXO1 activation in response to continuous H₂O₂ production

Bolus H₂O₂ treatment causes a rapid spike in cellular H₂O₂ concentration that is rapidly depleted from cells (less than two hours)[43,47]. To determine how continuous H₂O₂ exposure affects the timing of p53 and FOXO1 activation, we used glucose oxidase (GOX), an enzyme which uses glucose and O₂ as substrates to produce H₂O₂[47]. Glucose concentration is in excess in cell-culture media (~25 mM in our imaging media), and thus GOX produces H₂O₂ continuously over a 24-h period without significant depletion of glucose. Though the rate of H₂O₂ production is not constant and dips after the first hour of GOX exposure[48].

We treated MCF7 cells harboring FOXO1-mVenus, p53-mCherry and H2B-ECFP reporters to four different concentrations of GOX and measured p53 and FOXO1 activation every 20 min for 24 h. Like bolus H₂O₂ treatment, GOX showed dose-dependent accumulation of FOXO1 in the nucleus, with the proportion of cells with nuclear FOXO1 increasing with dose (Fig. 3a–e). Yet the pattern of FOXO1 nuclear shuttling is different. Bolus H₂O₂ treatment results in nuclear shuttling of FOXO1 within 1 h and the timing is largely independent of dose (Fig. 2c–e). In contrast, the time of FOXO1 nuclear accumulation after GOX treatment was dose-dependent, occurring earlier at higher doses likely due to more rapid accumulation of H₂O₂ (Fig. 3b–e). In addition, for bolus H₂O₂ treatment, the time in which FOXO1 remained nuclear increased with dose (Fig. 2c–e), yet after GOX exposure, once FOXO1 entered the nucleus it remained in the nucleus until the end of the experiment or until the cell died, in the majority of cells (Fig. 3b–e). We did observe a few exceptions in the .5 and .75mU/mL doses of GOX (see bottom left cell in Fig. 3a for example). Thus, clearance of H₂O₂, as occurs for bolus H₂O₂ exposure, is likely required for FOXO1 to shuttle back to the cytoplasm.

In contrast to bolus H₂O₂ treatment, p53 accumulation precedes nuclear FOXO1 accumulation under continuous H₂O₂ production by GOX (Fig. 3a–e). Interestingly, once FOXO1 enters the nucleus, p53 levels remain mostly constant, neither increasing nor decreasing. This can be visualized in single-cell heat maps sorted by the time in which FOXO1 entered the nucleus (Fig. 3b–e). Plotting the derivative of p53 accumulation sorted in this fashion (Δp53/hour plots, Fig. 3b–e), there is a spike in p53 accumulation (positive derivative) that precedes shuttling of FOXO1 to the nucleus, but rapidly decays to a derivative of ~0 (steady state) upon FOXO1 nuclear entry. Aligning trajectories to the time that FOXO1 entered the nucleus further highlights this (Fig. 3f, g). Prior to FOXO1 shuttling to the nucleus, p53 has a positive derivative while after FOXO1 shuttles to the nucleus the derivative declines to 0, suggesting p53 levels remain constant while FOXO1 is in the nucleus. Together these data show that under continuous external

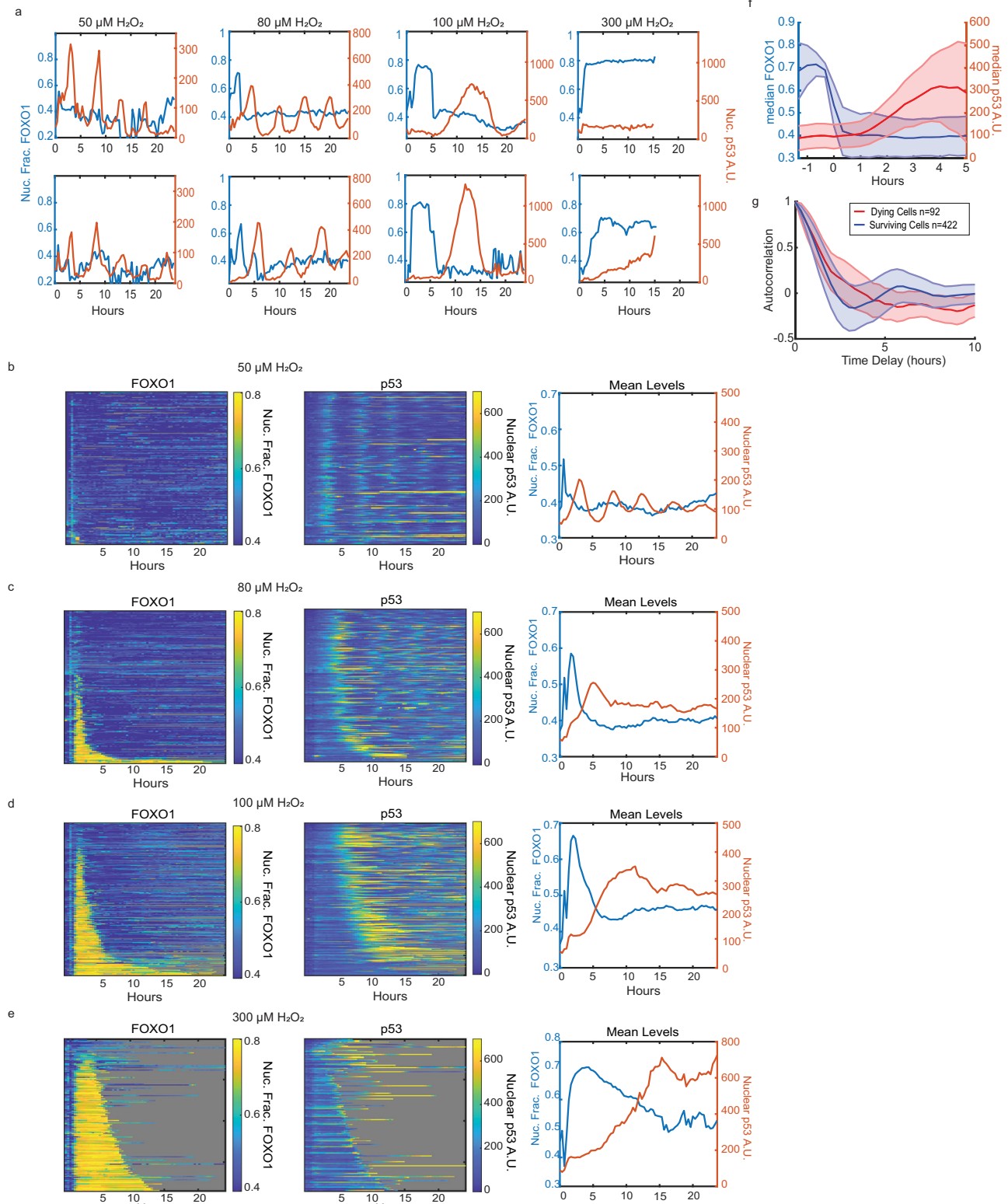

**Fig. 2 | FOXO1 activation precedes p53 activation at high concentration of acute H₂O₂ treatment. a** Representative single-cell traces of nuclear fraction of FOXO1 (blue, left y-axis) and nuclear levels of p53 (red, right y-axis) of cells treated with 50 μM, 80 μM, 100 μM and 300 μM of H₂O₂ for 24 h. **b–e** Heat maps of single-cell traces of nuclear fraction of FOXO1-mVenus (left), nuclear p53-mCherry (middle), and mean of both (right) for 24 h following H₂O₂. Each row of the heat maps is a single cell over time. Both FOXO1 and p53 heat maps are sorted by the duration that FOXO1-mVenus remained in the nucleus. Gray indicates cell death. **b** 50 μM

H₂O₂ (*n* = 188 cells, 1% cell death). **c** 80 μM H₂O₂ (*n* = 238 cells, 11% cell death). **d** 100 μM H₂O₂ (*n* = 288 cells, 34% cell death). **e** 300 μM H₂O₂ (*n* = 186 cells, 97% cell death). **f** Median levels of FOXO1 and p53 of all cells in 80 μM and 100 μM treatments with traces aligned to when FOXO1 exited the nucleus. **g** Autocorrelation of p53 trajectories of dying (red, *n* = 92) vs surviving (blue, *n* = 422) cells, treated with 80 μM and 100 μM of H₂O₂ from Figs. 2C and 2D. Shaded error bars in (**f**, **g**) represent the median absolute deviation. A.U. - Arbitrary Units. Nuc. Frac. - Nuclear Fraction. Source data are provided in Source Data Fig. 2.

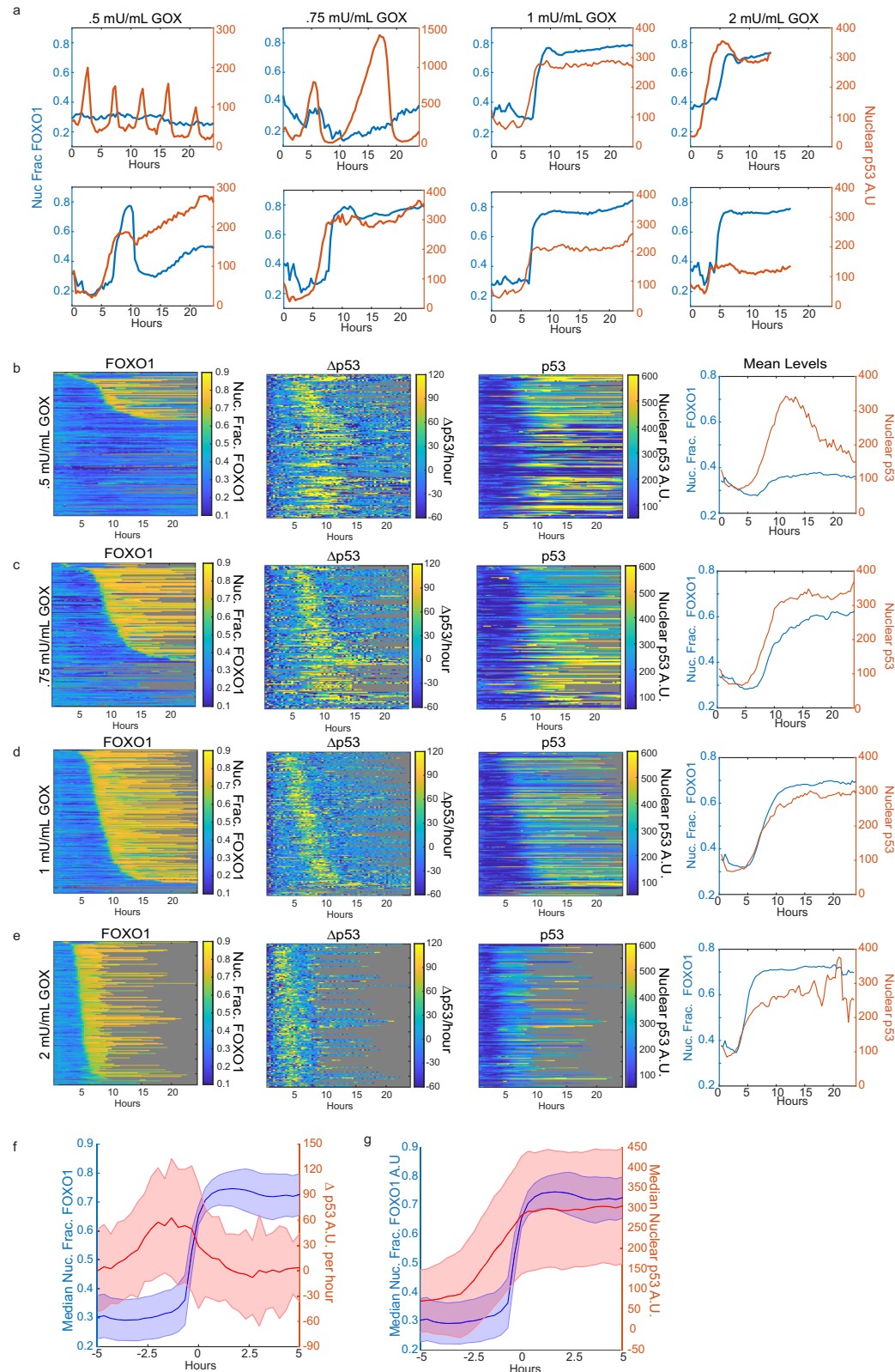

<blockquote>
production of $H_2O_2$ by GOX, p53 accumulates prior to FOXO1 nuclear shuttling. Yet once FOXO1 enters the nucleus p53 levels remain constant. For the remainder of this study, we focused on acute $H_2O_2$ stress as it provided clear differences in TF activation at fixed timepoints (Fig. 1), without the need for timelapse imaging. In the discussion, we speculate on why the order of FOXO1 and p53 activation differs between acute and continuous $H_2O_2$ stress.
</blockquote>

## Additional TFs are activated with either FOXO1 or p53 but not both, under acute $H_2O_2$ stress

Activation of p53 at low and FOXO1 at high $H_2O_2$ concentrations prompted us to ask whether other TFs are co-activated with p53 or FOXO1. To identify potential TFs, we treated MCF7 cells with PBS as a control and two different concentrations of $H_2O_2$ (50 μM and 75 μM), isolated individual nuclei, and performed single-cell Assay for

**Fig. 3 | p53 accumulation precedes FOXO1 activation under continuous H$_2$O$_2$ production. a** Representative single-cell traces of nuclear fraction of FOXO1-mVenus (blue, left y-axis) and nuclear levels of p53-mCherry (red, right y-axis) of cells treated with .5 mU/mL (milliunits per milliliter), .75 mU /mL, 1 mU/mL, and 2 mU/mL of glucose oxidase (GOX) for 24 h. **b**–**e** nuclear fraction of FOXO1-mVenus (left) the derivative or rate of p53-accumulation in A.U. per hour (2nd from left), Heat maps of single-cell traces for nuclear p53-mCherry (3rd from left) and mean of FOXO1-mVenus and p53-mCherry (right) for 24 h following GOX treatment. Each row of the heat maps is a single cell over time. All heat maps are sorted from top to bottom by the time in which FOXO1 entered the nucleus. Gray indicates cell death. **b** 0.5 mU/mL GOX ($n$ = 176 cells, 27% cell death). **c** 0.75 mU/mL GOX ($n$ = 154 cells, 50% cell death). **d** 1 mU/mL GOX ($n$ = 232 cells, 63% cell death). **e** 2 mU/mL GOX ($n$ = 195 cells, 97% cell death). **f** Median levels of FOXO1 and median rate of p53 accumulation of all cells treated with GOX. Traces are aligned to when FOXO1 entered the nucleus. **g** same as (**f**), but median p53 levels are plotted. Shaded error bars in (**f**, **g**) represent the median absolute deviation. A.U. - Arbitrary Units. Nuc. Frac. - Nuclear Fraction. Source data are provided in Source Data Fig. 3.

Transposase-Accessible Chromatin using sequencing (ATAC-seq) and gene expression using the 10X genomics single-cell Multiome kit. Unsupervised clustering of the ATAC-seq data identified six separate clusters (Fig. 4a, b). Clusters four and five represented PBS control nuclei, while the other clusters were observed predominantly in H$_2$O$_2$ treated nuclei. Clusters two and three were enriched with FOXO motifs but not p53 motifs, and we refer to these clusters together as the FOXO cluster (Supplementary Fig. 3A). In contrast, nuclei in cluster six were enriched for p53 motifs, but not FOXO motifs (Supplementary Fig. 3B).

We then focused on other transcription factor motifs that were enriched with either the FOXO or p53 cluster but not both (Supplementary Fig. 3A, B). Within the FOXO cluster, we observed an enrichment in HSF, GRHL1, NF-κB, ZKSCAN1 and NFAT TF binding motifs (Fig. 3c and Supplementary Fig. 3A). To determine if these TFs showed similar activation kinetics as FOXO1, as suggested by the ATAC data, we measured the NF-kB isoform RELA and NFAT isoform NFAT1 together with p53 in single cells by immunofluorescence five hours after H$_2$O$_2$ treatment (Fig. 4e–h). Similar to FOXO1, we observed mutually exclusive activation of RELA and NFAT1 with p53, with both TFs showing maximum activation at the highest concentration of H$_2$O$_2$ (Fig. 4f–h). In contrast, both NFAT1 and RELA were activated in FOXO1 active cells at high H$_2$O$_2$ concentrations (Supplementary Fig. 3E, F). RELA was only activated at the highest concentrations of H$_2$O$_2$ and only in a subset of cells consistent with the ATAC-seq data (Fig. 3c, Supplementary Fig. 3E). We also measured nuclear HSF1 levels but did not observe a change in concentration in response to H$_2$O$_2$ (Supplementary Fig. 3D). Cells in the FOXO cluster did have an increase in the expression of the canonical HSF target genes *HSPA1A* (log2 fold-change .98, $P < 10^{-5}$) and *HSPA1B* (log2 fold-change 1.1, $P < 10^{-6}$). Thus, the enrichment in HSF motifs might be due to upregulation of HSF2/HSF4, or regulation of these factors independent of an increase in nuclear levels as shown in previous studies[49].

The p53 cluster was enriched for motifs in the AP-1 family of TFs (JUN, FOS, and ATF subfamilies), as well as NRF2 (*NFE2L2* gene) (Fig. 4d and Supplementary Fig. 3B). The AP-1 family of TFs have similar binding motifs, which might give false positives for activation[50]. We picked two AP-1 TFs (JUN and FOS) and NRF2 to validate the ATAC-seq data. Like p53, the nuclear levels of JUN and NRF2 rose at low H$_2$O$_2$ concentrations and reverted to baseline at higher concentrations, coinciding with FOXO1 activation (Fig. 4i–l, Supplementary Fig. 3G, H). FOS on the other hand was activated by H$_2$O$_2$ in cells with both nuclear FOXO1 and cytoplasmic FOXO1 (Supplementary Fig. 3C).

Finally, we tested the generality of these findings by measuring co-activation of p53 with RELA, NRF2, and JUN in MCF10A cells by immunofluorescence (Supplementary Fig. 3I–K). Similar to MCF7 cells, p53, and RELA activation is mutually exclusive, while p53 is co-activated with JUN and NRF2. Together these data suggest that other H$_2$O$_2$ activated TFs are upregulated either with FOXO1 (RELA, NFAT1) or p53 (JUN, NRF2), but not both, following H$_2$O$_2$ treatment.

## The role of the peroxiredoxin/sulfiredoxin system in controlling the switch between p53 and FOXO1 activation

Next, we investigated the mechanism underlying the switch from activating p53 at low H$_2$O$_2$ concentrations, to activating FOXO1 at high H$_2$O$_2$ concentrations. Since other TFs are activated with either FOXO1 or p53, with distinct mechanisms of control, we reasoned that the mechanism is likely upstream of the direct regulators of each TF. A redox relay stemming from a 2-Cys PRDX protein is a plausible mechanism as redox relays can affect multiple proteins and are switched off by hyperoxidation when H$_2$O$_2$ levels cross a particular threshold, as described below.

2-Cys PRDX proteins function as homodimers and harbor two key cysteine residues: a peroxidatic cysteine (Cp), whose thiol group is oxidized to sulfenic acid (SOH) by H$_2$O$_2$, and a resolving cysteine (Cr), which reacts with the sulfenic acid form of Cp in trans, forming a disulfide bond between the two monomers (Fig. 5a)[51]. This disulfide bond can be resolved by the thioredoxin system, or it can participate in a redox relay, where it transfers oxidative equivalents to downstream proteins, altering their function (the Cp sulfenic acid can also transfer oxidative equivalents)[52]. However, at high levels of H$_2$O$_2$, the Cp of PRDX is hyperoxidized from sulfenic acid to sulfinic acid (SO$_2$H), or to sulfonic acid (SO$_3$H), rendering it incapable of forming a disulfide bond with Cr or participating in redox relays (Fig. 5a). The SO$_2$H and SO$_3$H states are referred to as hyperoxidation. We hypothesized that the switch from p53 accumulation at low H$_2$O$_2$, to FOXO1 activation at high H$_2$O$_2$ is due to hyperoxidation of one or more PRDX proteins. Switching off FOXO1 and the subsequent activation of p53 would only occur after SRXN1 repairs hyperoxidized PRDX, similar to Pap1 activation in yeast (Fig. 5a)[24,25].

To test this hypothesis, we first measured hyperoxidation of 2-Cys PRDX proteins following H$_2$O$_2$ treatment by western blot. We observed dose-dependent hyperoxidation at concentrations in which cells activated FOXO1 in previous experiments (Fig. 5b). We next employed non-reducing gels to assess the hyperoxidation of PRDX1 and PRDX2 isoforms (Fig. 5c, d). During cell lysis, 2-Cys PRDX proteins undergo oxidation and subsequently form intermolecular disulfide bonds, which are visible as dimers on non-reducing gels[53]. In contrast, hyperoxidized PRDX proteins, which cannot form these bonds, are detected as monomers. We again observed dose-dependent hyperoxidation of PRDX1 and PRDX2, at concentrations in which cells activated FOXO1 as opposed to p53 in prior experiments. PRDX2 was hyperoxidized at slightly lower doses than PRDX1 as shown in a previous study[54]. We then tested the effect of conoidin A, an inhibitor of PRDX1 and PRDX2 (Supplementary Fig. 4A–D)[55,56]. Treatment of cells with 5 μM conoidin A resulted in the activation of FOXO1 and prevented p53 accumulation at low doses of H$_2$O$_2$ (Supplementary Fig. 4B), consistent with a role for hyperoxidation and inactivation of PRDX1 and/or PRDX2 as key events in activating FOXO1 and repressing p53 activation. Conoidin A had a similar effect in MCF10A cells (Supplementary Fig. 4E, F). Furthermore, in MCF7 cells, 10 μM conoidin A resulted in FOXO1 activation but not p53 activation in the absence of H$_2$O$_2$ (Supplementary Fig. 4D). Taken together, these data suggest that inactivation of 2-Cys PRDX proteins, in particular PRDX1 and/or PRDX2, is associated with FOXO1 activation and suppression of p53 activation in response to H$_2$O$_2$.

We next tested the role of PRDX1 and PRDX2. Previous studies indicated that PRDX1 forms disulfide bonds with FOXO3, and PRDX1 depletion enhances nuclear FOXO3 upon H$_2$O$_2$ exposure[30,31]. We generated a CRISPR/Cas9-mediated *PRDX1* knockout (*PRDX* KO) line

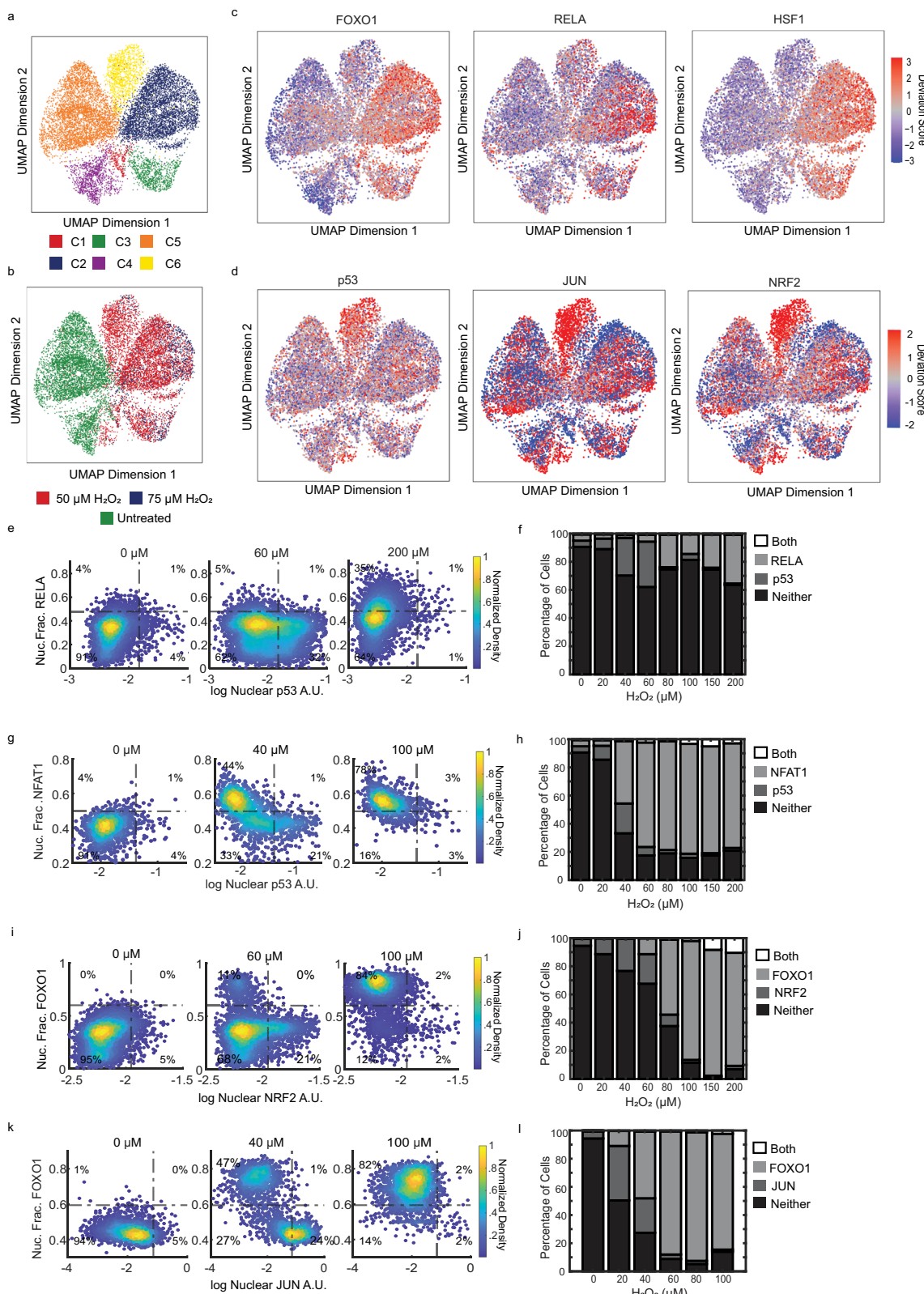

(Supplementary Fig. 4H). In *PRDX1* KO cells, there is a substantial increase in FOXO1 active cells and a subsequent decrease in p53 active cells at $H_2O_2$ concentrations ≥40 μM and above compared to controls (Fig. 5e–h). There is an increase in p53 activation at 20 μM $H_2O_2$ in *PRDX1* KO cells compared to controls. This is likely due to an increase in DNA damage, as PRDX1 is required for proper repair of DNA DSBs[57]. Consistent with this, DNA damage, as measured by γH2AX, was

significantly increased in *PRDX1* KO cells following $H_2O_2$ treatment (Supplementary Fig. 4I). We were unable to make a CRISPR knockout line of *PRDX2*, so instead took a doxycycline inducible shRNA approach (Supplementary Fig. 4J). Similar to *PRDX1* KO cells, *PRDX2* knockdown led to increased FOXO1 activation at lower $H_2O_2$ concentrations (Supplementary Fig. 4K–N). Together these data suggest that PRDX1 and PRDX2 inhibit FOXO1 activation in response to $H_2O_2$ stress, and

**Fig. 4 | Additional H$_2$O$_2$ induced transcription factors are activated with either FOXO1 or p53. a** Uniform manifold approximation and projection (UMAP) plot of cells treated with PBS, 50 μM and 75 μM of H$_2$O$_2$ after unsupervised clustering ($n \geq 10,000$). Colors for cells based on the six clusters obtained. **b** UMAP of the same cells in A but colored based on sample. **c** UMAPs of cells from A colored by deviation scores for FOXO1 (left), RelA (middle) and HSF1 motifs (right) (**d**) UMAPs of cells from A showing deviation scores for p53 (left), Jun(middle) and NRF2 motifs (right) (**e**) Density colored scatter plots of log nuclear p53 (x-axis) and nuclear fraction of RelA (y-axis) at indicated levels of H$_2$O$_2$ treatment for 5 h (**f**) Percentage of cells activating both RelA and p53 (Both), RelA only, p53 only or neither for all concentrations of H$_2$O$_2$ (**g**) Density colored scatter plots of log nuclear p53 (x-axis) and nuclear fraction of NFAT1 (y-axis) at indicated levels of H$_2$O$_2$ treatment for 5 h (**h**) Percentage of cells activating both NFAT1 and p53 (Both), NFAT1 only, p53 only or neither for all concentrations of H$_2$O$_2$. **i** Density colored scatter plots of log nuclear NRF2 (x-axis) and nuclear fraction of FOXO1(y-axis) at indicated levels of H$_2$O$_2$ treatment for 5 h (**j**) Percentage of cells activating both FOXO1 and NRF2 (Both), FOXO1 only, NRF2 only or neither for all concentrations of H$_2$O$_2$. **k** Density colored scatter plots of log nuclear JUN (x-axis) and nuclear fraction of FOXO1 (y-axis) at indicated levels of H$_2$O$_2$ treatment for 5 h (**l**) Percentage of cells activating both FOXO1 and JUN (Both), FOXO1 only, JUN only or neither for all concentrations of H$_2$O$_2$. A.U. - Arbitrary Units. Nuc. Frac. - Nuclear Fraction. Source data are provided in Source Data Fig. 4.

their inactivation through hyperoxidation may drive the switch between p53 and FOXO1 activation.

To test the role of hyperoxidation in activating FOXO1 and inhibiting p53, we sought to block repair of hyperoxidized PRDX proteins by SRXN1. To do this we used J14, a small molecule inhibitor of SRXN1. At concentrations of H$_2$O$_2$ 40 μM and above, J14 led to a decrease in p53 active cells, and a corresponding increase in FOXO1 activation, similar to *PRDX1* KO cells (Fig. 4i, j). J14 treatment had a similar effect in MCF10A cells (Supplementary Fig. 4G). To confirm that the effect of J14 was due to inhibition of SRXN1 and not due to off-target effects, we knocked down *SRXN1* using shRNA and observed a similar shift from p53 activation to FOXO1 activation (Supplementary Fig. 5A–D). In addition, live-cell microscopy revealed that J14 prolonged FOXO1 activation and delayed p53 activation, supporting a role for SRXN1 in shutting off FOXO1 and activation of p53 (Supplementary Fig. 5B). Prolonged FOXO1 activation by J14 was accompanied by increased cell death (14% H$_2$O$_2$ alone vs 70% H$_2$O$_2$ + J14, Supplementary Fig. 5B).

If hyperoxidation of PRDX1/2 is required for activating FOXO1 and repressing p53, then overexpressing *SRXN1* should raise the H$_2$O$_2$ threshold for FOXO1 activation. To test this, we established a cell line (SRXN1-OE) in which *SRXN1* is expressed from the PGK promoter and confirmed reduced PRDX hyperoxidation by western blot (Supplementary Fig. 5E). SRXN1-OE cells show a striking decrease in FOXO1 activity at high concentrations of H$_2$O$_2$ (80–150 μM) and a corresponding increase in p53 activation (compare Fig. 5k, l to e, f). The effect of *SRXN1* overexpression was reversed by treatment with the SRXN1 inhibitor, J14 (Supplementary Fig. 5F). Together these data support a model where PRDX1 and PRDX2 inhibit FOXO1 activation and facilitate p53 activation; inactivation of PRDX1 and PRDX2 by hyperoxidation shifts the balance from p53 to FOXO1 activation at elevated H$_2$O$_2$ levels; and reactivating p53 while turning off FOXO1 involves SRXN1-mediated repair of hyperoxidized PRDX1/2. We address potential limitations of this model in the discussion section.

### Evidence that p53 is activated by a hyperoxidation event that occurs at low H$_2$O$_2$ concentrations

As noted above, SRXN1-OE cells activated p53 and inhibited FOXO1, at much higher levels of H$_2$O$_2$ (80–150 μM) than control cells. However, at lower H$_2$O$_2$ levels (20–60 μM), SRXN1-OE cells showed fewer p53 active cells than control cells (Fig. 5k, l). This suggests that a hyperoxidation event, which occurs at low H$_2$O$_2$ levels, is likely required for p53 activation in some cells. In support of this, inhibition of SRXN1 with J14 caused an increase in p53 active cells at 20 μM H$_2$O$_2$ (Fig. 5j).

### The two temporal phases of transcription factor activation cause distinct transcriptional changes

We next asked what gene expression changes occur in the two TF phases in response to H$_2$O$_2$ using RNA-seq. To identify genes activated in each phase, we used three different treatment groups, MCF7 cells, MCF7 cells treated with the SRXN1 inhibitor J14, and MCF7 cells overexpressing SRXN1. We performed RNA-seq on each group with and without 50 μM H$_2$O$_2$ as this concentration led to a mix of FOXO1 and p53 active cells with H$_2$O$_2$ alone, mostly FOXO1 active cells in the J14 treatment group (phase 1), and mostly p53 active cells in the SRXN1 overexpression group (phase 2). Log2 fold changes (L2FC) and *p* values were calculated by comparing H$_2$O$_2$ treated samples to their relevant PBS controls.

A heatmap of differentially expressed genes revealed broad differences in gene expression between J14 and SRXN1-OE samples (Fig. 6a). Gene Set Enrichment Analysis (GSEA) using a list of known p53[58] and NRF2[59] target genes verified that these genes were enriched in SRXN1-OE cells when compared to J14 treated cells (Supplementary Fig. 6A–F, adjusted $P < 1*10^{-9}$). Upregulated genes in the p53 and NRF2 pathway included genes in DNA repair (*XPC, POLH, DDB2, ASCC3*), pentose phosphate pathway/nucleotide biosynthesis (*TIGAR, RRM2B, MTHFD2*), NADPH production (*ALDH3A1, ME1*) and glutathione synthesis (*GCLM, GCLC, SLC7A11*). Similarly, a volcano plot comparing gene expression in the SRXN1-OE cells to J14 treated cells revealed that SRXN1-OE cells show significant increases in p53 (*CDKN1A, MDM2, SESN2*) and NRF2 (*SLC7A11, TXNRD1, HMOX1*) target genes (Fig. 6b). While the J14 treated samples show significant increases in heat shock protein genes (*HSPA1A, HSPA1B*).

We next focused on the differences between the H$_2$O$_2$ response in SRXN1-OE and J14 treated cells. GSEA using the hallmark (H), curated gene sets (C2), and regulatory gene sets (C3) from the Molecular Signatures Database revealed that J14 treated cells, show significant upregulation of ribosome, proteasome, oxidative phosphorylation, and spliceosome genes when compared to SRXN1-OE cells (Fig. 6c, d). Together these data show that the different groups of TFs activated in each phase result in different gene expression patterns. In phase 1, cells upregulate critical components for protein production (ribosomes, spliceosome), protein quality control (proteasome, heat shock proteins), and oxidative phosphorylation. In contrast during phase 2, cells upregulate expression of genes in DNA repair, the pentose phosphate pathway, nucleotide biosynthesis, NADPH production, and glutathione biosynthesis.

## Discussion

Excess levels of H$_2$O$_2$ activate a diverse set of TFs, including p53, NRF2, JUN, FOXO, NF-κB, and NFAT1 that act to restore the cellular redox environment and repair cell damage induced by H$_2$O$_2$. Here we found that which TFs are activated, and their timing of activation are dependent on H$_2$O$_2$ dose and the rate of exposure (Fig. 7).

Measuring TF activation in single cells early after acute H$_2$O$_2$ stress revealed the dose dependent nature of TF activation (Fig. 4). At low levels of H$_2$O$_2$ stress, cells accumulate p53, NRF2 and JUN (p53 group) while FOXO1, NF-κB and NFAT1 (FOXO1 group) remain inactive. In contrast under high levels of H$_2$O$_2$ stress this is reversed; the FOXO1 group of TFs are activated, while accumulation of the p53 group is blocked. This suggests there are separate thresholds of H$_2$O$_2$ required to activate each group of TFs with lower H$_2$O$_2$ levels required to activate the p53 group and higher levels required to activate the FOXO1 group.

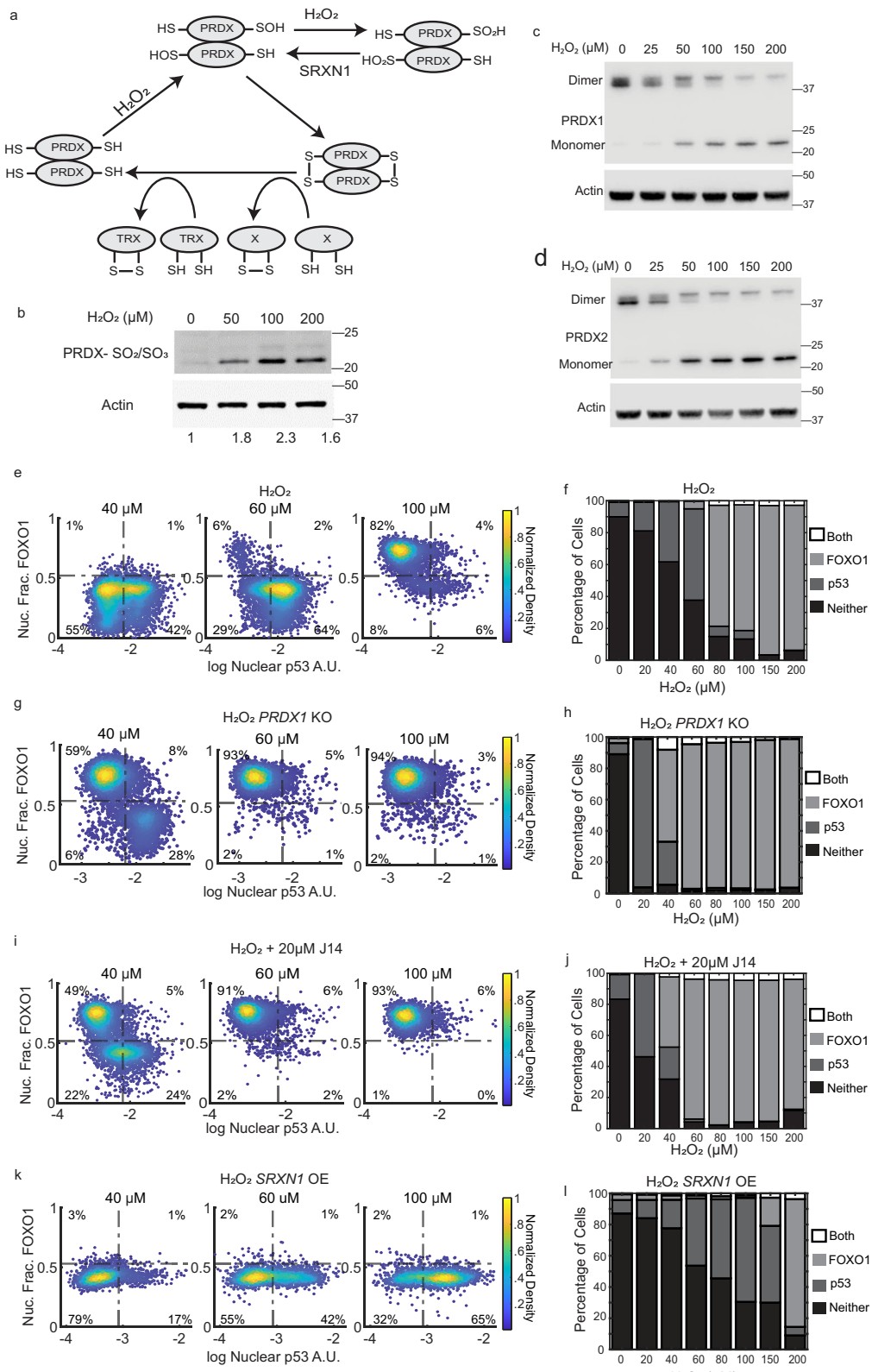

We found evidence that PRDX1 and PRDX2 inactivation by hyperoxidation is a key event and likely the mechanistic basis for the switch from activation of the p53 group of TFs to activation of the FOXO1 group (Fig. 5). Knockout of *PRDX1* and knockdown of *PRDX2* lowered the concentration of $H_2O_2$ required to switch from p53 to FOXO1 activation. Furthermore, overexpression of *SRXN1* reduced the hyperoxidation of PRDX proteins (Supplementary Fig. 5E), and increased the

concentration of $H_2O_2$ required to switch from p53 to FOXO1 activation (Fig. 5k, l). One caveat of these data is that other proteins outside the PRDX family are hyperoxidized by $H_2O_2$, and are repaired by SRXN1, and we cannot rule out that these PRDX independent hyperoxidation events are involved in activation of p53 and FOXO1[60].

How inactivation of peroxiredoxins directly regulates the TFs in this study is not known. Many of these TFs, or their upstream

**Fig. 5 | The role of the Peroxiredoxin/Sulfiredoxin system in controlling the switch between p53 and FOXO1 activation. a** Schematic of the redox cycle of PRDXs. $H_2O_2$ oxidizes the peroxidatic cysteine to sulfenic acid (SOH), which can form a disulfide bond in trans with the resolving cysteine. At high $H_2O_2$ concentrations, the peroxidatic cysteine is further oxidized to $SO_2H$. PRDX-$SO_2H$ can be repaired by SRXN1 to sulfenic acid. Oxidized peroxiredoxins can be reduced by the Thioredoxin system or can transfer oxidative equivalents to other proteins as depicted by protein X in the diagram. **b** Western Blot stained for hyperoxidized ($SO_2/SO_3$) PRDX1/2/3/4 and Actin in MCF7 cells treated with indicated concentrations of $H_2O_2$ for 3 h. Experiment was one of 3 biological replicates with similar results. Non-reducing western blots of (**c**) PRDX1 and (**d**) PRDX2, exposed to

different concentrations of $H_2O_2$. Actin was used as a loading control. Experiments are one of two biological replicates with similar results. Density colored scatter plots (**e, g, i, k**) and percentage of cells activating both FOXO1 and p53, only FOXO1, only p53 or neither (**f, h, j, l**). **e, f** Cells treated with $H_2O_2$ used as a control at the indicated concentrations of $H_2O_2$ for 5 h. **g, h** PRDX1 knockout cells treated with $H_2O_2$ at indicated concentrations for 5 h. **i, j** Cells treated with 20 μM of J14, an inhibitor of SRXN1, along with $H_2O_2$ at the indicated concentrations for 5 h. **k, l** Cells overexpressing SRXN1 (*SRXN1* OE) treated with indicated concentrations of $H_2O_2$ for 5 h. A.U. - Arbitrary Units. Nuc. Frac. - Nuclear Fraction. Source data are provided in Source Data Fig. 5.

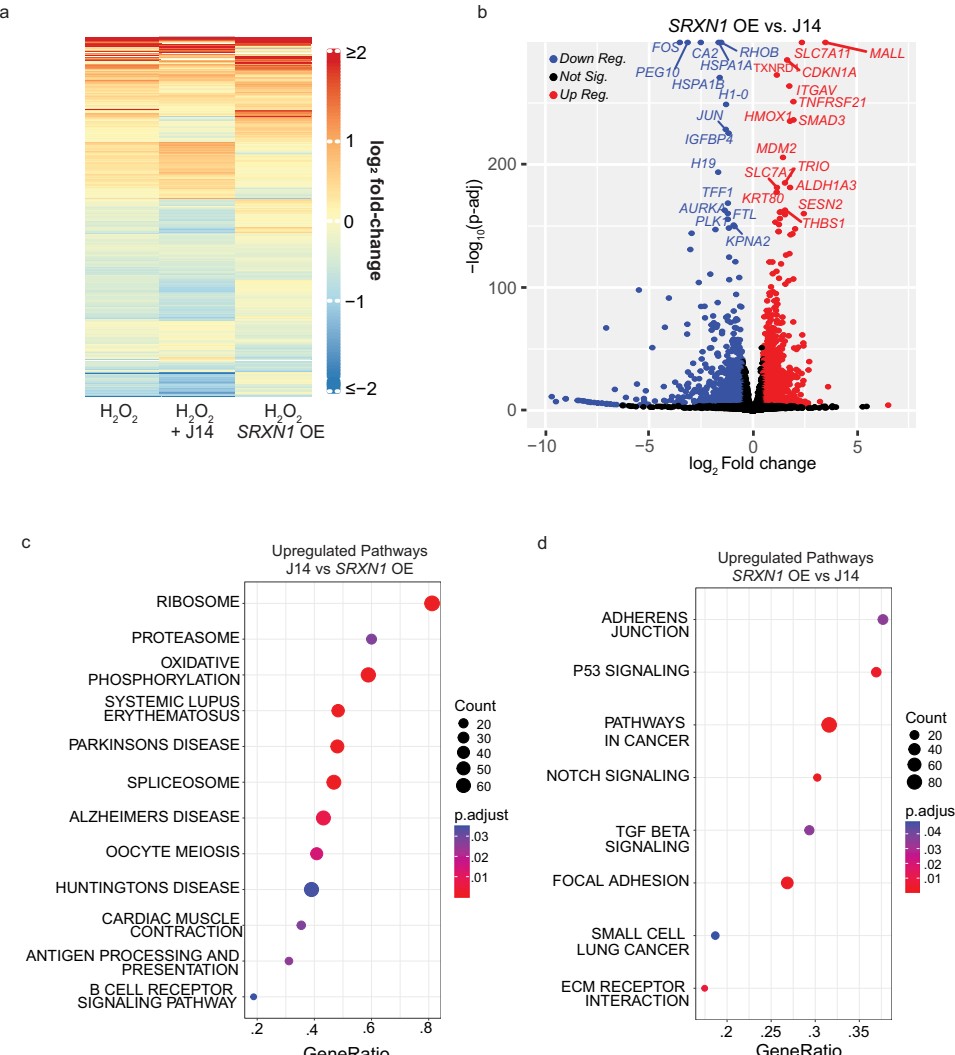

**Fig. 6 | The two temporal phases of transcription factor activation cause distinct transcriptional changes. a** Significantly up and downregulated genes ($P < 0.0001$, Wald test, |Log2Fold Change| > 0.2) in $H_2O_2$ treated cells vs. PBS controls, $H_2O_2 + J14$ treated cells vs J14 treated controls and $H_2O_2$ treated *SRXN1* OE cells vs. *SRXN1* OE PBS controls. $H_2O_2$ concentration is 50 μM, J14 20 μM. **b** Volcano plot showing log2 fold-change and -log10 adjusted *p* value (Wald test) of *SRXN1* OE cells

treated with $H_2O_2$ vs J14 treated with $H_2O_2$. **c** GSEA of KEGG pathways upregulated in cells treated with J14 + $H_2O_2$ as compared to SRXN1-OE + $H_2O_2$. **d** GSEA of KEGG pathways upregulated in *SRXN1* OE cells treated with $H_2O_2$ as compared to cells treated with J14 + $H_2O_2$. *SRXN1* OE - *SRXN1* overexpression. *P* values in (**c, d**) were calculated by permutation test and adjusted using the Benjamini-Hochberg correction.

regulators, harbor reactive cysteines that regulate their activity. For example, KEAP1 regulates NRF2 by sequestering it in the cytoplasm and targeting it for ubiquitination by the BTB-CUL3-RBX1 E3 ubiquitin ligase complex[61]. $H_2O_2$ treatment leads to a disulfide bond between two cysteines in KEAP1 that prevent NRF2 degradation[62]. Activation of p53 in response to $H_2O_2$ has been linked to ATM, p38 and JNK kinase activity and all three kinases are known to be activated by disulfide

bonds either on the protein itself (ATM) or in upstream proteins which regulate kinase activity (p38, JNK)[16,63–66]. Similarly, FOXO3 has been shown to form disulfide bonds with PRDX1 which sequester FOXO3 in the cytoplasm[30,31]. In addition, AKT, which phosphorylates FOXO proteins causing cytoplasmic sequestration, is inhibited by disulfide bond formation between two cysteines in the protein, which leads to its inactivation[67]. In future studies, it will be interesting to determine

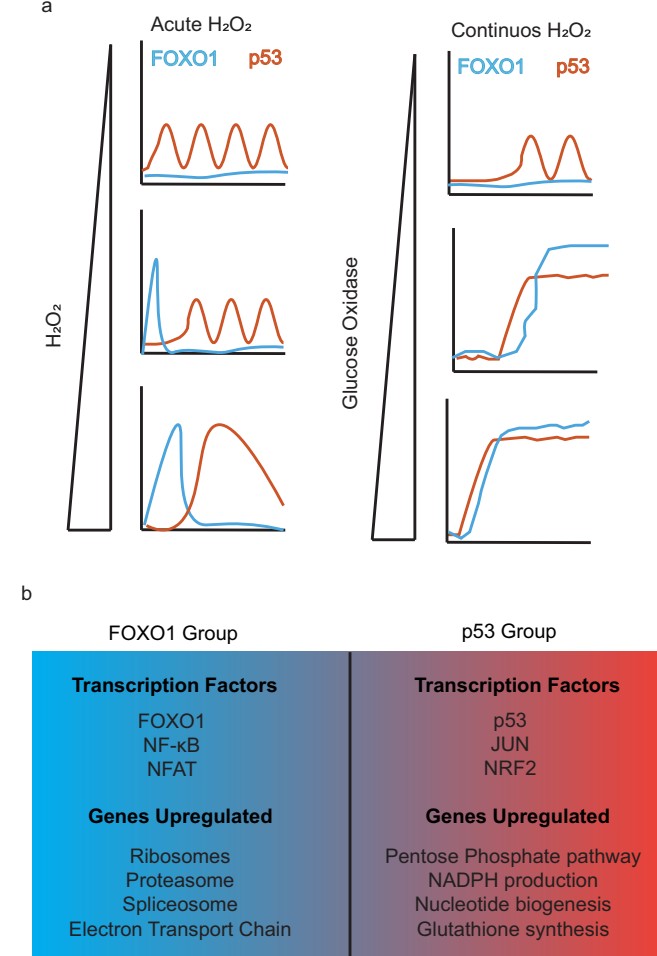

**a**

Acute H₂O₂
FOXO1    p53

Continuos H₂O₂
FOXO1    p53

H₂O₂

Glucose Oxidase

**b**

| FOXO1 Group | p53 Group |
|---|---|
| **Transcription Factors** | **Transcription Factors** |
| FOXO1 | p53 |
| NF-κB | JUN |
| NFAT | NRF2 |
| **Genes Upregulated** | **Genes Upregulated** |
| Ribosomes | Pentose Phosphate pathway |
| Proteasome | NADPH production |
| Spliceosome | Nucleotide biogenesis |
| Electron Transport Chain | Glutathione synthesis |

**Fig. 7 | Temporal activation of transcription factors by H₂O₂ stress. a** Model showing early FOXO1 activation followed by p53 activation in response to acute H₂O₂ stress (left). Lower levels of H₂O₂ induce p53 oscillations whereas higher levels of H₂O₂ cause p53 to become more sustained in cells. The duration of FOXO1 increases as the dose of H₂O₂ is increased. In response to continuous H₂O₂ from glucose oxidase (right), p53 accumulates before FOXO1 but stops accumulating once FOXO1 is active. **b** Different transcription factors are activated with FOXO1 (left) and p53 (right). The two transcription factor groups upregulate very distinct transcriptional programs.

whether particular cysteine oxidation events require a specific PRDX protein and how this is impacted by PRDX hyperoxidation.

At high levels of H₂O₂ there are two distinct temporal phases of TF activation, and the order of TF activation is dependent upon the mode of H₂O₂ delivery (Fig. 7a). Under a high level of acute H₂O₂ stress, FOXO1 is activated by shuttling to the nucleus, while p53 levels remain low. In the second phase, FOXO1 switches off by shuttling to the cytoplasm and p53 begins to accumulate. In contrast, under continuous production of H₂O₂ from glucose oxidase, the order of TF activation is reversed with p53 accumulation occurring before FOXO1 activation. Once FOXO1 is activated, p53 levels remain constant, neither increasing nor decreasing.

Our working model to explain the difference between acute and continuous H₂O₂ stress, is that there is likely a threshold level of cellular H₂O₂ required to activate p53, and a separate higher threshold required to activate FOXO1 (and block p53 accumulation). In response to high levels of acute H₂O₂ stress (and menadione treatment), both thresholds are crossed nearly simultaneously, and thus FOXO1 accumulates in the nucleus and p53 levels remain close to baseline. In contrast, under continuous production of H₂O₂ by GOX, there is a

temporal delay between when H₂O₂ concentrations reach the lower threshold required for p53 activation and when concentrations exceed the threshold required to activate FOXO1 and inhibit p53 accumulation. Hence p53 levels accumulate prior to FOXO1 accumulation under continuous H₂O₂ production.

One pitfall of our study is that we have not explored TF activation in response to a purely intracellular source of ROS. Menadione exposure results in mutually exclusive activation of FOXO1 and p53 similar to H₂O₂ (Supplementary Fig. 1K). This suggests that H₂O₂ produced within the cell elicits a similar TF response as extracellular H₂O₂ exposure as menadione induces ROS through redox cycling, and menadione induced cell death is suppressed by intracellular catalase expression[41,42]. However, we cannot exclude the potential for menadione producing ROS through reactions with the media. Future studies using d-amino acid oxidase enzymes localized to specific cellular compartments will be useful for determining whether intracellular H₂O₂ behaves similar to extracellular H₂O₂ exposure, and if the TF response is dependent on the location of H₂O₂ production within the cell.

Using RNA-seq, we found that the two groups of TFs upregulate different target genes, suggesting there are likely distinct transcriptional changes required for handling the different levels of H₂O₂ stress. We would like to stress that since many different TFs are activated with the p53 group of TFs and the FOXO1 group of TFs, we do not know which genes are upregulated by any particular TF. The p53 group of TFs activate genes in NADPH and glutathione synthesis, and nucleotide production all of which are key mechanisms of combating oxidative stress. In contrast, when the FOXO1 group of TFs are activated under high H₂O₂ stress, there is an increase in expression of ribosome, proteasome, heat shock proteins and components of the electron transport chain (ETC). Higher levels of H₂O₂ are likely indicative of damage to critical cellular components. For example, ribosomal RNA and proteins are damaged by oxidative stress and restoring ribosomes might be necessary under high levels of H₂O₂[68]. In a similar fashion, repairing cellular damage caused by H₂O₂ stress is energy intensive and upregulating ETC genes might be required to ensure adequate energy production. Furthermore, damage to the ETC can result in an increase in H₂O₂[69]. Thus, directly adding H₂O₂ to cells might recreate the signal that the ETC is damaged, without damaging components of the ETC.

## Methods

### Cell lines
MCF7 cells were a gift from Galit Lahav, Harvard Medical School and were validated by short tandem repeat profiling by the University of Arizona Genetics Core in 2019. A549 (CCL-185), U-2 OS (HTB-96) and MCF10A (CRL-10317) cells were obtained from ATCC and were validated by ATCC using short tandem, repeat profiling in 2018. All cell lines were tested free of mycoplasma by DAPI stain.

### Cell culture
MCF7 cells were grown in Roswell Park Memorial Institute 1640 medium (RPMI) supplemented with 10% FBS, 100 units/mL penicillin, 100 μg/mL streptomycin, and 25 ng/mL amphotericin B. A549 (CCL-185) and U-2 OS (HTB-96) were grown in Dulbecco's Modified Eagle Medium (DMEM) supplemented with the same concentrations of FBS and antibiotics as mentioned above. MCF10A (CRL-10317) were grown in DMEM/F-12 (Invitrogen #11330-032) media supplemented with 5% Horse serum (Invitrogen#16050-122), EGF (20 ng/mL final), Hydrocortisone (0.5 mg/mL final), Cholera Toxin (100 ng/mL final), Insulin (10 μg/mL final), 1% Pen/Strep (100x solution, Invitrogen #15070-063).

### Cell treatments
For H₂O₂ treatments, H₂O₂ was diluted in PBS, then added directly to the media to get the final concentrations indicated in each experiment.

The stock of $H_2O_2$ (Fisher Scientific AAL13235AP) was replaced monthly. For glucose oxidase (GO) treatments (Sigma, G7141-50KU), the desired concentration of the enzyme was made in sodium acetate (JTBaker, 3460-01) buffer (50 mM) and added directly to the media. For J14 (MedChemExpress, HY-135008), a stock of 10 mM was created in DMSO and added directly to the media for a final concentration of 20 μM. A stock concentration of 35 mM of Conoidin A (Cayman Chemical Item no. 15605) was made in DMSO. The stock was diluted in media and then added to cells to obtain the indicated final concentration. Tert-butyl hydroperoxide (TBHP, 70% solution in water) was obtained from Acros Organics and diluted in PBS. The TBHP PBS stock was then added directly to media to obtain the indicated final concentration. A stock solution of Menadione (Sigma Aldrich M5625) was made in 100% Ethanol, diluted in media, and then added to cells to obtain the final concentration indicated for each experiment.

## Plasmid and cell line construction

Lentivirus was produced and infection was carried out as described by ref. 70. On day 1, $5 * 10^6$ HEK 293 T cells were plated into 10 cm dishes in DMEM + 10% FBS. After 24 h, media was replaced with 12 mL of fresh DMEM + 10% FBS. The transfection reagent was prepared by adding 18 μl TransIT Transfection Reagent LT1 Cat# MIR 2304. We then added 3.2 μg of the lentiviral plasmid, 1.8 μg of pMDLg/pRRE (Addgene: 12251), .7 μg of pRSV-Rev (Addgene: 12253) and .3 μg of pMD2.G (Addgene: 12259). The mix was incubated at room temperature for 30 min and then added to the 293 T cells. On Day 4 the media was harvested and stored in a 50 mL falcon tube and stored overnight at 4 °C. 5 mL of fresh DMEM + 10% FBS was added. On day 5 the remaining media was harvested to the falcon tube. This was then spun down at 800 g for 20 min and the filtered through a .45 μM filter. Virus was stored at −80 °C in 1 ml aliquots.

For lentiviral infection we plated $.5 * 10^5$ cells on day one. After 24 h we prepared lentivirus by adding protamine sulfate to a final concentration of 8 μg/mL and 1 μl of 1 M HEPES to 1 mL of thawed virus. Media was then aspirated from cells and replaced with 1 mL of virus and another 1 mL of fresh cell culture media. Cells were incubated at 37 °C for 4–6 h and then viral media was replaced with 10 mL of fresh tissue culture media.

To construct the p53-mCherry reporter, we used multisite gateway cloning (Invitrogen, Eugene, OR). The human ubiquitin promoter was cloned upstream of p53 and mCherry was cloned downstream of p53 in a lentiviral destination vector harboring the puromycin resistance gene. Plasmids were sequenced for verification. The SRXN1 overexpression vector was designed and constructed using VectorBuilder (https://en.vectorbuilder.com/). The vector is a lentiviral vector that expresses mCerulean-NLS-P2A-T2A-SRXN1 from the PGK promoter and harbors a blasticidin resistance gene. The NLS (nuclear localization sequence) tagged mCerulean is separated from SRXN1 with the P2A-T2A self-cleavage site so that SRXN1 is independent of the nuclear mCerulean signal. The mCerulean signal allows the verification that the construct is expressed in cells. MCF7 cells were infected with the lentivirus, cells were selected using blasticidin and clones were isolated and validated. The PRDX1 knockout line was made by using a CRISPR/Cas9 lentiviral vector from VectorBuilder (hPRDX1[gRNA#1077]). MCF7 cells were infected with lentivirus and selected using blasticidin. Individual clones were isolated and validated using Western Blot. SRXN1 knockdown cells were made by using plasmid from Vector Builder (pLV[shRNA]-Puro-U6 > hSRXN1[shRNA#1]) to transfect MCF7 cells. Cells were then selected using puromycin and the knockdown was validated using Immunofluorescence. CRISPR/Cas9 was used to tag the endogenous locus of FOXO1 at the C-terminus with the mVenus fluorescent protein in MCF10A cells using the eFlut system described previously[71]. We then added a lentiviral H2B-ECFP tag described previously and the p53-mCherry reporter was then added to the FOXO1 tagged MCF10A cell line for experiments in Supplementary Fig. 2[43]. The doxycycline inducible shPRDX2 vector was obtained from Horizon Discovery (TRIPZ, clone_Id: V3THS_380088). For this vector 2nd generation lentiviral vectors were used to make lentivirus.

## Immunofluorescence

Cells (1700 cells/well) were plated in glass bottom 96 multi-well plates (CellVis) or polystyrene plates from PerkinElmer (CellCarrier-96 Cat# 6055300). Cells were allowed to attach for two days and treated at different time points. Cells were fixed with 2% PFA for 10 min, permeabilized using 0.1% Triton X-100 in PBS for 10 min, blocked with 2% BSA in PBS and incubated o/n at 4 °C in primary antibodies made with 2% BSA and 0.1% Tween in PBS. The cells were then washed two times with PBS followed by incubation with secondary antibodies at room temperature for 1 h. The cells were then washed two times in PBS followed by staining with DAPI and imaged in PBS. Images were analyzed Cell Profiler[72]. To obtain cytoplasmic levels of FOXO1, a ring of 3 pixels wide was drawn around the nuclear mask and mean cytoplasmic FOXO1 was extracted using this mask. All plots were made using MATLAB. Primary antibodies used: Anti-FOXO1 (C29H4) from Cell Signaling Cat# 2880S (1:500), Anti-p53 (DO-1) from Santa Cruz Cat# sc-126 (1:500), Anti-Sulfiredoxin from Santa Cruz Cat# sc-166786 (1:100), NRF2 (D1Z9C) XP from Cell Signaling Rabbit mAb #12721 (1:500), NFAT1 (D43B1) XP from Cell Signaling Rabbit mAb #5861 (1:300), NF-kB p65 (D14E12) XP from Cell Signaling Rabbit mAb #8242 (1:400), Anti-HSF1 antibody 10H8 from Stress Marq Biosciences (1:200), c-Fos (9F6) from Cell Signaling Rabbit mAb # 2250 (1:1000), and c-Jun (60A8) from Cell Signaling Rabbit mAb #9165 (1:400).

Secondary antibodies used for IF: Goat Anti-Rabbit IgG (H + L) Alexa Fluor 488 Cat# A-11034, Goat Anti- Mouse IgG (H + L) Alexa Fluor 594 Cat#A-11032, Goat Anti-Rabbit IgG (H + L) ALexa Fluor 546 Cat# A-11010, Goat Anti- Mouse IgG (H + L) Alexa Fluor 647 Cat# A21236 all obtained from Invitrogen. All secondary antibodies were used at a concentration of 1:500.

Cells were imaged on a Nikon Eclipse Ti-E microscope. Data was acquired using the NIS Elements AR 5.21.02 software and visualized with NIS Elements Viewer 5.21. DAPI was imaged using the AT-DAPI Filter Set (Chroma) for 40–60 ms, Alexa Fluor 488 was imaged using (Chroma) AT-EGFP/F Filter Set for 600–800 ms, Alexa Fluor 594 was imaged using (Chroma) AT-TR/mCH Filter Set for 600–800 ms, Alexa Fluor 546 was imaged using (Chroma) AT-TRITC/CY3 filter set and Alex Fluor 647 was imaged using AT-CY5.5 filter set.

For protein oxidation experiments (Supplementary Fig. 1J), reactive thiol groups and total proteins were stained using as described previously[39]. MCF7 cells were treated with different concentrations of $H_2O_2$ for 3 h and then fixed, permeabilized, blocked, and incubated with 100 nM of Alexa Fluor 488 C5-maleimide (Invitrogen, A10254) in PBS for 10 min at room temperature. Total protein was stained using 2 μM of Alexa Fluor 647 NHS ester (succinimidyl ester) (Invitrogen, A37573) in PBS for 10 min at room temperature.

For lipid peroxidation experiments (Supplementary Fig. 1K), MCF7 cells were treated with different concentrations of $H_2O_2$ and fixed, permeabilized, blocked, and incubated with 4-Hydroxynonenal monoclonal antibody (1:50) (Thermo Fisher Scientific, MA5-27570) overnight at 4 °C. The cells were then washed with PBS and incubated for an hour at room temperature with Alexa Fluor 594. DAPI was used to stain the nuclei. Cells treated with 100 μM of erastin (APExBIO, cat. #B1524) for 24 h was used as a positive control.

Fluorescence intensity of staining was quantified using CellProfiler version 3.19. To extract the nuclear intensities, we segmented nuclei using DAPI. For the cytoplasmic signal a ring 2 pixels in diameter around the nucleus was used. Data analysis and plots were performed in MATLAB version 2021a.

## Live cell microscopy

Cells (15,000 cells/well) were plated on 12 well glass bottom plates (CellVis) which were coated with poly L-lysine (Sigma) and allowed to attach for 48 h. The cells were grown in the appropriate media as mentioned above and then rinsed with PBS and given DMEM Fluorobrite (ThermoFisher) media with 2% FBS,100 units/mL penicillin, 100 μg/mL streptomycin, 25 ng/mL amphotericin B, and 1x Glutamax (ThermoFisher). Cells were imaged every 15–20 min for 24–48 h by a Nikon Eclipse Ti-E microscope. Data was acquired using the NIS Elements AR 5.21.02 software. A thin layer of mineral oil was used to prevent evaporation. Temperature (37 °C) and 5% $CO_2$ levels were maintained using the OKO labs incubation system. H2B-CFP was imaged using the C-FL AT ECFP/Cerulean Filter Set (Chroma) for 20–40 ms. FOXO1–mVenus was imaged using (Chroma) ET-EYFP Filter Set for 600–800 ms, p53-mCherry was imaged using (Chroma) AT-TR/mCH Filter Set for 600–800 ms. Movies were analyzed using p53Cinema[73].

## Autocorrelation analysis

Autocorrelation analysis in Fig. 2g and Supplementary Fig. 2A–D was performed using the autocorr function in MATLAB. For each cell, the autocorrelation was performed on a 10-h window of the p53 traces. For cells that did not accumulate FOXO1 in the nucleus, the autocorrelation was performed on the first 10 h. For cells in which FOXO1 shuttled to the nucleus, autocorrelation was performed on the 10 h after FOXO1 exited the nucleus.

## Comet assay

Comet assays were performed as described previously[74]. $4 \times 10^4$ cells per well were plated to 6 well plates and grown for 24 h. Cells were then aspirated, washed once with PBS, trypsinized, and resuspended in cell culture media. Cells were pelleted by centrifugation (200 g for 5 min), media was then aspirated, and cells were with washed 1x PBS. Cells were pelleted again by centrifugation (200 g for 5 min) and resuspended in ice cold PBS to a final concentration of $2 \times 10^4$ cells/mL. Cells were then treated with different concentrations of $H_2O_2$, NCS, or PBS as a control and incubated on ice for 20 min. Next cells were mixed with 1% low melting point agarose in distilled water at a ratio of 2 parts cell mixture to 1 part agarose and incubated at 37 °C. 50 μl of the agarose/cell suspension was pipetted onto Comet Slides (R&D systems Cat. 4250-200-03) and cooled at 4 °C for 10 min to solidify. The slides were immersed in prechilled (4 °C) lysis solution (R&D systems Cat. 4250-050-01) overnight at 4 °C. Excess buffer was drained the next day. For the alkaline comet assays, slides were immersed in freshly prepared alkaline solution 0.03 M NaOH, 2 mM $Na_2$EDTA (pH > 13) and incubated in the dark at room temperature for 1 h. Electrophoresis was performed on slides at 16 volts for 30 min at 4 °C. For neutral comet assays, slides were immersed in 50 mL of TBE buffer for 5 min and electrophoresis was performed in TBE buffer at 16 volts for 10 min at 4 °C. After electrophoresis, slides were immersed in dd$H_2O$ twice for 10 min, immersed in 70% ethanol for 5 min, and dried at 37 °C for 30 min. 50 μl of 10 μg/mL propidium iodide solution was added to each well of the slide followed by imaging on a microscope.

## Western blots

Cells (approx.100,000 cells/dish) were plated to a 6 cm dish and incubated for 48 h. They were washed with PBS, scraped off the plate, centrifuged and the cell pellet was lysed using a Lysis Buffer (25 mM Tris pH 7.6, 150 mM NaCl, 1% NP-40, 1% Na-deoxycholate, and 0.1% SDS in water + protease inhibitor cocktail [Sigma] + phosphatase inhibitor cocktail [Sigma-Aldrich] + okadaic acid + sodium fluoride). The cells were spun down and the supernatant was used to measure protein concentration by using a Bradford Assay (BioRad). Equal protein concentrations were loaded onto a NuPAGE 4-12% Bis-Tris gels (Invitrogen). Protein was transferred to a nitrocellulose membrane and incubated in blocking solution (PBS, 5% BSA, 0.1% Tween 20) for 1 h at room temperature. The membrane was incubated with primary antibodies at 4 ºC overnight, rinsed three times with PBST and incubated with secondary antibodies for 1 h at room temperature. The blot was imaged on the LI-COR Odyssey. Primary Antibodies used: Anti-Peroxiredoxin-SO3 antibody (ab16830) (1:1000), Recombinant Anti-Peroxiredoxin 1/PAG antibody [EPR5433] (ab109498) (1:1000) from Abcam and actin Cat#A2228 clone AC-74 (Sigma) was used for primary staining. Secondary antibodies used: 680LT secondary LICOR-IR Dye Cat# 925-68020, and 800CW secondary LICOR-IR Dye Cat# 925-3221, both at 1:10000 concentration.

## Non-Reducing Western Blots

Approximately 100,000 cells were plated to 6 cm dishes and incubated for 48 h. Media was aspirated and replaced with 2 mL fresh RPMI. Dilutions of $H_2O_2$ in PBS were added directly to this media at the indicated concentrations for 2 h. Cells were washed with PBS, scraped off the plate, centrifuged, and the cell pellet was lysed using lysis Buffer (25 mM Tris pH 7.6, 150 mM NaCl, 1% NP-40, 1% Na-deoxycholate, and 0.1% SDS in water + protease inhibitor cocktail [Sigma]). Protein concentration was measured using Pierce BCA Assay (Thermo). Samples were prepared using 20 μg of protein with LDS Non-Reducing Sample Buffer 4X (Thermo) without heat denaturation or reductant added unless indicated. These were loaded onto a NuPAGE 4–12% Bis-Tris gel (Invitrogen). Protein was transferred onto a PVDF membrane using the iBlot 2 Gel Transfer Device and incubated in blocking solution (PBS, 5% NFDM, 0.1% Tween 20) for 1 h at room temperature. The membrane was incubated with primary antibodies at 4 °C overnight, rinsed three times with PBST and incubated with secondary antibodies for 1 h at room temperature. SuperSignal West Pico PLUS Chemiluminescent Substrate (Thermo) was used to image the membranes on a Licor Odyssey Fc imager. Primary antibodies used: Recombinant Anti-Peroxiredoxin 1/PAG antibody [EPR5433] (ab109498) (1:2000), and Recombinant Anti-Peroxiredoxin 2/PRP antibody [EPR5154] (ab109367) (1:3000) from Abcam and A2228 Monoclonal Anti-β-Actin antibody (1:5000) from Sigma. Secondary antibodies used: Mouse Anti-rabbit IgG-HRP: sc-2357 (1:25,000) from Santa Cruz Biotechnology and LICOR-IR Dye 800CW Cat# 925-3221 secondary (1:10,000).

## Single Cell ATAC and RNA sequencing

MCF7 cells were plated (50,000 cells/well) on plastic 6 well plates and allowed to attach for 48 h. Cells were then treated with PBS control, 50 μM $H_2O_2$, and 75 μM $H_2O_2$ for 5 h. Nuclei were then isolated using the Nuclei Isolation for Single Cell Multiome ATAC + Gene Expression protocol (CG000365). All further steps with the kit were performed by the University of Arizona Genetics Core. Sequencing was carried out using Novogene. Chromium Next GEM Single Cell Multiome ATAC + Gene Expression kit (Product code: 1000285) from 10X Genomics was used for transposition, GEM generation and Barcoding, ATAC Library Construction, cDNA Amplification, Gene expression and library construction (CG000338).

## Single cell sequencing analysis

Fragment files generated via 10x Genomics were analyzed using the ArchR pipeline version 1.01[75]. Untreated cells and those treated with 50 μM and 75 μM $H_2O_2$ were separated into six clusters using an iterative latent semantic indexing algorithm acting on tiles of 500-bp for each cell using the ArchR function IterativeLSI for dimensionality reduction and Seurat's findClusters with a resolution of 0.14. Peaks were called in each cluster using MACS2 using the addReproduciblePeakSet function in ArchR. The getMarkerFeatures function was used to generate lists of relevant features from each of clusters 2, 3, and 6 with clusters with high numbers of untreated cells used as background groups (clusters 1, 4, and 5). Each cluster was analyzed to assess the prevalence of relevant transcription factors with

JASPAR2020 binding profiles. Motif enrichment was evaluated on the generated features for each cluster for motifs with FDR < 0.01 and | LFC | > 1.3 using the peakAnnoEnrichment function in ArchR. Chrom-VAR deviations for each transcription factor were evaluated on a per-cell basis[76]. Transcription factors were ordered by average ChromVAR deviations in each cluster and compared to z-scores of expression levels for each gene from scRNAseq data.

## Bulk RNA sequencing

MCF7 cells (50,000 cells/well) were plated on plastic 6 well plates and allowed to attach for 2 days. There were six different treatment groups: MCF7 + PBS Control, MCF7 + 50 μM H2O2, MCF7 + 20 μM J14 + PBS Control, MCF7 + 20 μM J14 + 50 μM H2O2, MCF7 SFRX-OE + PBS Control, MCF7 SFRX-OE + 50 μM H2O2. SFRX-OE indicates the cell line harboring a construct that expressed the human Sulfiredoxin gene from the PGK promoter. J14 is a Sulfiredoxin inhibitor. Five hours after treatment, RNA was isolated using RNeasy Mini Kit (Cat. No. 74104) from Qiagen. The isolated RNA was then sent to Novogene for sequencing.

## Bulk differential expression analysis

Analysis was performed by importing BAM files with the *Rsubread* package and *featureCounts* program dropping all genes with counts less than 10 across experimental conditions[77]. We then performed differential expression analysis on the dataset using *DEseq2* version 1.38.3[78]. J14 treated, SRXN1-OE, and wild-type samples were compared between 50 μM H2O2 and no treatment. A list of genes was generated for each set of replicates from these comparisons using thresholds after effect size shrinkage using the *apeglm* package[79]. These lists were combined to evaluate overall expression changes across each treatment condition. We also evaluated expression differences that were of opposite direction across the J14 and sulfiredoxin conditions. We compared differential expression of untreated sulfiredoxin and J14 samples with their wild-type counterparts to get an estimate of expression differences from these conditions independent of hydrogen peroxide and confirmed these trends using Principal Component Analysis.

## Bulk gene set enrichment analysis

Gene Set Enrichment Analysis was performed to evaluate differences in sulfiredoxin and J14 samples treated with 50 μM H2O2 using the clusterProfiler R package version 4.6.2[80] and functions from the enrichplot package (*enrichplot: Visualization of Functional Enrichment Result*, version 1.18.4). R package version 1.18.3, https://yulab-smu.top/biomedical-knowledge-mining-book/). We included KEGG gene sets and custom gene sets of p53 and NRF2 target genes. The list of p53 target genes was obtained from Fischer, 2017[58] and included p53 targets with direct regulation in seven different studies. NRF2 target genes were obtained from ref. 59.

## Statistics and reproducibility

We did not employ any statistical methods to predetermine sample size. The experiments were not randomized and the investigators were not blinded to the source of the experimental samples and outcome assessment.

All immunofluorescence experiments were replicated a minimum of three times with reproducible results. Time lapse movies were run at least three times. As long as cell number was maintained the data are reproducible as the impact of H2O2 exposure decreases with cell number. Single cell ATAC sequencing was carried out once as the cost of the experiments for cost reasons. However key findings of the single-cell ATAC data were validated using immunofluorescence. Bulk RNA sequencing was performed in triplicate for each condition. Western blots were replicated at least twice for each experiment.

For time-lapse microscopy heat maps in Figs. 2 and 3, cells with high p53 levels before H2O2 treatment were excluded from the heat maps for clarity but included in all other analysis. These were less than 5% of cells.

## Reporting summary

Further information on research design is available in the Nature Portfolio Reporting Summary linked to this article.

## Data availability

Data from single-cell Assay for Transposase-Accessible Chromatin using sequencing (ATAC-seq) and gene expression using the 10X genomics single-cell Multiome kit, as well as bulk RNA-seq data is available on NCBI's Gene Expression Omnibus (GEO) database accession number: GSE227556. Other source data for graphs in the Figures and Supplementary Figs. is provided in the Source Data Files. Source data are provided with this paper.

## Code availability

All custom written scripts described in the manuscript are available upon request.

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

## Acknowledgements
We thank members of the Paek lab, the Thorne lab, J. Stewart-Ornstein and S. Chen for helpful comments and discussion. We thank A. Capaldi and T. Weinert for feedback on the manuscript. We also thank the University of Arizona Genetics Core for performing the Single cell ATAC and Gene expression protocol. This work was supported by National Institutes of Health Grants RO1GM130864 and NIH T32-GM132008 (W.M.-S.). B.A.W. received support from the Arnold & Mabel Beckman Foundation Scholars Program.

## Author contributions
E.J. and A.L.P. designed the research. E.J. performed and/or contributed to all experiments. W.M-S. and A.L.P. analyzed all the data from the Single Cell ATAC sequencing and Bulk RNA sequencing. B.A.W. contributed to validation of other transcription factors involved in this study, and performed non-reducing western blots. L.S. and B.A.W. contributed to making the fluorescent tagged lines and knockout lines. C.P. contributed to Fig. 1h. I.A.C. and J.B.S. performed comet assays in Fig. 1. E.J., W.M.S., B.A.W., C.P., and A.L.P. performed data analysis. E.J. and A.L.P. wrote the manuscript. All authors have reviewed and commented on the manuscript.

## Competing interests
The authors declare no competing interests.
