## [Peer Review File · Nature Communications]

Temporal Coordination of the Transcription Factor Response to H₂O₂ stressREVIEWER COMMENTS

Reviewer #1 (Remarks to the Author):

In the study "temporal coordination of the transcription factor response to H₂O₂ Stress," Jose et al. I. propose that H₂O₂-responsive transcription factor (TF) activation is tightly co-ordinated over time in an H₂O₂-dose dependent manner. The authors examined FOXO1 and p53 nuclear translocation in MCF-7 cell lines in response to different doses of H₂O₂. The authors also identified other TF co-activated with either FOXO1 or p53. Based on nuclear translocation kinetics, the authors conclude a biphasic TF response that is H₂O₂-dose dependent. The concept of the study is intriguing.

However, there are some general concerns. Much emphasis was placed on identifying other TFs that might also be regulated biphasic by H₂O₂ before considering the fundamental question of TF targets in response to H₂O₂. In other words, FOXOs and p53 play roles in proliferation, apoptosis, senescence, DNA damage, etc., and the TF's selection of the targeted promoter depend on posttranslational modifications on the TF. Even disregarding this point, the length of treatment is a significant component that contributes to target selection. Redox signaling is complex, depending on time and dose, localization of H₂O₂ production, etc. Considering this, exogenous H₂O₂ treatment is ok as a starting point, but from a biological point of view, not rigorous enough to conclude a biphasic response of TFs to H₂O₂. Thus, studies including localized H₂O₂ production in the cell must be considered.

Another concern is the reductionist approach of analyzing only one cell line in depth without confirming the bi-phasic phenotype in other cell lines, including the above-discussed results (target validation; H₂O₂ timing).

Other points:

Phosphorylated γ H2AX analysis as proof of DNA damage: very superficial analysis. What type of DNA damage? Is the γ H2AX bound to damaged DNA? Importantly, H₂O₂ has been shown to induce γ H2AX phosphorylation independent of DNA damage. That also speaks to the point above: What are the p53 targets under these conditions?

Targets of FOXO1 and p53 need to be examined in an H₂O₂-time and dose-dependent manner.

Regarding the PRDX1 FOXO3a interaction and FOXO1 regulation by PRDX1: PRDX1 was identified to interact via inter disulfides with FOXO3a C31 and C150. FOXO3a is the only FOXO member with a C150 residue. Experiments addressing the regulation of FOXO1 by PRDX1 are not shown.

PRDX biology: The authors show a PRDXSO3 immunoblot. There are six mammalian PRDXs, four of which are 2 -cysteine PRDXs, and they should interact with this antibody. Which isoform of PRDX shown in Figure 4 is being examined needs to be clarified.

Also, PRDX redox regulation by H₂O₂ includes dimer formation. Mono-sulfinic/sulfonic oxidized PRDX proteins may exist as dimers. Careful titration curves need to be shown documenting monomeric, dimeric, and overoxidized PRDX on non-reducing gels.

This goes back to a point just mentioned above. Sulfiredoxin has other clients besides PRDX1. It also nitrosylated PRDX2. Which PRDX is responsible for FOXO1 translocation/activation?

The H₂O₂ doses used are very high, unphysiologically high, for cell lines shown in the supplemental data.

Reviewer #2 (Remarks to the Author):

In this work, Jose and cols have explored the temporal coordination of transcription response to H₂O₂ stress. By using cutting-edge technologies, such as live cell microscopy, single cell ATAC and RNA sequencing, authors find a differential coordination along the time of the transcriptional program in response to different doses of H₂O₂. Authors propose that low concentrations of H₂O₂ mainly activate p53, whereas high concentrations induce 2 different phases: in a first phase, FOXO1, in the second phase, p53. Other transcription factors are activated in the first phase with FOXO1 or in the second phase with p53, but not with both. Finally, authors provide evidence about the role of peroxiredoxins in the control of this transcriptional program.

Work is of great interest to better understand the cell response to H₂O₂ but it shows different weaknesses that must be considered before publication of the manuscript.

MAJOR POINTS

1. One of the main concerns with this work is that the use of different concentrations of acute treatment with H₂O₂ also incorporates a variable, which is the time in which the H₂O₂ is acting, since H₂O₂ will disappear, either spontaneously, or by the cellular catalase. Indeed, the time for action of 40-50 uM H₂O₂ may be much shorter than the time in which the 200 uM H₂O₂ is acting. Did the authors analyze the decrease in the H₂O₂ concentrations in the culture medium along the time? The main results obtained at the beginning of the work, particularly the differential timing and intensity in the regulation of FOXO1 and p53, must be repeated by using a continuous system of H₂O₂ production, such as the combination of xanthine-xanthine oxidase (or other enzymatic continuous production of H₂O₂), where controlling the xanthine oxidase concentration the level of H₂O₂ will change, and this level will be maintained along the time (with enough xanthine as substrate). If results would not be identical, the focus of the work would be the temporal coordination of the transcription factor response to acute H₂O₂ stress. This experiment is even more relevant when looking at the results with ter-butyl-hydroperoxide, a mimetic of the H₂O₂ that is maintained in a constant concentration in the culture due to the inability of the catalase to metabolize it. Results with this compound are significantly different.

2. Most of the work (with the exception of a punctual experiment at the beginning of the work) has been made in only one cell line, which is a tumoral cell. Do authors consider that other cells, particularly non tumoral cells, will respond in the same way? In order to better evaluate the relevance of the work, at least some specific important results presented (such as the additional H₂O₂ induced transcription factors activated with either FOXO1 or p53 or the role of the peroxiredoxin/sulfiredoxin system) must be demonstrated in a non-tumoral cell, such as the MCF10A used in Fig S1.

3. The strong increase in nuclear p53 presented in Fig. 1A (40 uM H₂O₂) cannot be observed in the video Supplemental Movie 1. Stopping the movie at 5 h the image is more similar to that presented for untreated cells. There is no clear correlation, neither, in the 80 uM or 100 uM, when stopping the movie at 5 h. The only result that clearly correlates is the 200 uM concentration.

4. The only functional analysis of oxidation of intracellular components in this work is γ H2AX, which only refers to DNA damage and which was monitored for comparison with the situation of p53 activation. Do different doses of H₂O₂, or different times, affect differentially to oxidation of other components, such as proteins or lipids?

MINOR POINTS

1. From the beginning, authors focus on FOXO1 and p53 because both are activated by H₂O₂ regulating cell-cycle arrest and apoptosis. References for this assessment must be incorporated in the Introduction section. And not only FOXO1 or p53 are activated by H₂O₂ to regulate proliferation and cell death. Authors must better justify why to initiate the work focusing on these transcription factors.

2. Authors mention in the discussion section that: “.. pentose phosphate pathway and nucleotide biosynthesis in the second transcription factor phase.....Upregulation of these genes are likely important for completing DNA replication and DNA repair”. Pentose phosphate cycle will be relevant for NADPH production, additionally to ALDH3A1, ME1 that authors indicate.

Reviewer #3 (Remarks to the Author):

The manuscript “Temporal coordination of the transcription factors response to H₂O₂ stress” described how activation of different transcription factors is tightly and temporally coordinated in response to H₂O₂. The authors provided compelling evidence that p53 and FOXO1 are differentially activated in response to different levels of H₂O₂. Low levels of H₂O₂ activate p53 but not FOXO1, while high levels of H₂O₂ lead to two phases of transcription factor activation. FOXO1 is activated in the first phase, followed by p53 activation in the second phase in a H₂O₂ dose dependent manner. They further showed that other transcription factors (RelA, NFAT1, NRF2 and JUN) are also activated coordinately and differentially with FOXO1 and p53 in response to different levels of H₂O₂. The two temporal phases are associated with distinct gene expression changes corresponding to the changes in transcription factors. Lastly, they provided evidence that PRDX dependent redox relay dictates the activation of FOXO1 and p53, suggesting that peroxiredoxins control the type and timing of transcription factor activation. A similar PRDX-dependent redox relay mechanism exists in yeast, and now this study provided evidence for such mechanism in mammalian cells in response to H₂O₂. The temporal and coordinated activation of different transcription factors in response to different levels of H₂O₂ leads to gene expression changes that are necessary for the cytoprotective processes, providing a mechanistic understanding. Overall, the studies were well designed, methodologies are sound and detailed, and the data were strong in support of their conclusions.

There are a few concerns that need to be addressed.

1. What defines low vs. high H₂O₂ dose? Low dose used in MCF10A, A549, U2OS cells (200-300 μ M) is the high dose in MCF7. What causes the discrepancy? Should the intracellular H₂O₂ level be measured to define low vs. high dose? The doses used were quite high, causing significant cell death (~98% with 300 μ M), which may complicate the interpretation.
2. Is activated p53, JUN, or NRF2 in the same cells? Same question for FOXO1, RelA and NFAT1. ATAC-seq cannot distinguish each individual cell. From Figure 2, p53 is activated in ~70% of cells at 40 μ M, ~60% at 60 μ M, but nuclear Jun is in ~25% at 40 μ M and nuclear NRF2 is in ~20% of cells at 60 μ M, suggesting that these transcription factors may be activated in different cells. Measuring nuclear p53 and JUN/NRF2 simultaneously will answer this question, and same is true for the FOXO1/NFAT1 group. Answers to these questions will strengthen the transcriptomic data for the coordinated gene expression changes.
3. Figure 2 and Figure S2: autocorrelation assay is used but lacks the explanation how it is done in Materials and Methods.

REVIEWER COMMENTS

Reviewer #1 (Remarks to the Author):

In the study "temporal coordination of the transcription factor response to H₂O₂ Stress," Jose et al. I. propose that H₂O₂-responsive transcription factor (TF) activation is tightly coordinated over time in an H₂O₂-dose dependent manner. The authors examined FOXO1 and p53 nuclear translocation in MCF-7 cell lines in response to different doses of H₂O₂. The authors also identified other TF co-activated with either FOXO1 or p53. Based on nuclear translocation kinetics, the authors conclude a biphasic TF response that is H₂O₂-dose dependent. The concept of the study is intriguing.

However, there are some general concerns. Much emphasis was placed on identifying other TFs that might also be regulated biphasic by H₂O₂ before considering the fundamental question of TF targets in response to H₂O₂. In other words, FOXOs and p53 play roles in proliferation, apoptosis, senescence, DNA damage, etc., and the TF's selection of the targeted promoter depend on posttranslational modifications on the TF. Even disregarding this point, the length of treatment is a significant component that contributes to target selection. Redox signaling is complex, depending on time and dose, localization of H₂O₂ production, etc. Considering this, exogenous H₂O₂ treatment is ok as a starting point, but from a biological point of view, not rigorous enough to conclude a biphasic response of TFs to H₂O₂. Thus, studies including localized H₂O₂ production in the cell must be considered. Another concern is the reductionist approach of analyzing only one cell line in depth without confirming the bi-phasic phenotype in other cell lines, including the above-discussed results (target validation; H₂O₂ timing).

We thank reviewer #1 for their comments and helpful suggestions. We break down our response to each point for clarity.

The reviewer stated: "Much emphasis was placed on identifying other TFs that might also be regulated biphasic by H₂O₂ before considering the fundamental question of TF targets in response to H₂O₂. In other words, FOXOs and p53 play roles in proliferation, apoptosis, senescence, DNA damage, etc., and the TF's selection of the targeted promoter depend on posttranslational modifications on the TF."

We agree that identifying the TF targets in response to H₂O₂ would be valuable. This study is indeed a first step in determining which TFs are activated at different H₂O₂ concentrations and a broad look at the differences in gene expression programs (Figure 6) between the two different phases of the response. Identifying the specific transcriptional targets for each TF is a complex task and in our opinion beyond the scope of this study. For example, there are multiple FOXO transcription factors

activated by H₂O₂ and the genes activated by each FOXO isoform are likely redundant making knockout studies technically challenging. In addition, we observe different temporal patterns of p53 accumulation that are dose dependent (Figure 2). Previous studies have shown that the dynamics of p53 accumulation impact both gene expression programs and cell outcomes (Cell-cycle arrest, senescence, apoptosis) See: Purvis JE, Karhohs KW, Mock C, Batchelor E, Loewer A and Lahav G. (2012) "p53 dynamics control cell fate." Science, 336, 6087, Pp. 1440-1444.

Paek AL, Liu JC, Loewer A, Forrester WC and Lahav G. (2016) "Cell-to-Cell Variation in p53 Dynamics Leads to Fractional Killing." Cell, 165, 3, Pp. 631-642.

Thus the targets of a particular transcription factor likely change with the dose in complex ways. Again this is not to state that identifying specific targets is not important, but it is not the focus of the current study.

The reviewer further states: "Considering this, exogenous H₂O₂ treatment is ok as a starting point, but from a biological point of view, not rigorous enough to conclude a biphasic response of TFs to H₂O₂. Thus, studies including localized H₂O₂ production in the cell must be considered."

We agree that measuring p53 and FOXO1 activation by localized H₂O₂ production would be interesting, however we believe that this is beyond the scope of this study. We attempted to get the D-amino acid oxidase (DAAO) constructs targeted to the mitochondria, cytoplasm and nucleus working in our cells but were unable to obtain good transfection efficiency with preexisting constructs. We plan on constructing lentiviral DAAO constructs to explore this in future studies. Given that exogenous H₂O₂ is first imported through the cytoplasm, it is likely that at a minimum the cytoplasmic produced H₂O₂ would result in a similar biphasic response. But we agree that testing whether H₂O₂ produced in the mitochondria and nucleus lead to a similar response would certainly be interesting and is a goal of future studies.

We note that based on reviewer #2's suggestion we tested the p53 and FOXO1 response to continuous H₂O₂ production from glucose oxidase (GOX, see figure 3). Interestingly, we found that in response to continuous H₂O₂ production there is still temporal ordering in the activation of p53 and FOXO1. Yet under continuous H₂O₂ the order of activation was reversed; p53 was activated first, and then after a delay FOXO1 was activated. Interestingly, When FOXO1 was activated under continuous H₂O₂ production, p53 levels neither increased nor decreased but remained flat and elevated from baseline. The description of this is provided in the results section titled "p53 accumulation precedes FOXO1 activation in response to continuous H₂O₂ production"

lines 172-204. In the discussion, lines 381-388 we speculate on why the order of activation for p53 and FOXO1 differs between acute and continuous H₂O₂ production.

Next the reviewer states: “Another concern is the reductionist approach of analyzing only one cell line in depth without confirming the bi-phasic phenotype in other cell lines, including the above-discussed results (target validation; H₂O₂ timing).”

We have now included more data to show that the biphasic response we characterized is not limited to MCF7 cells. We focused on MCF10A, a non-cancerous breast epithelial cell line.

First, we created an MCF10A FOXO1-Venus, p53-mCherry cell line and performed time lapse imaging under acute H₂O₂ stress. We observed a dose-dependent increase in the duration of nuclear FOXO1 and delay of p53 activation until FOXO1 exits the nucleus (Supplemental Figures 2I-Q).

Second, using immunofluorescence in MCF10A cells, we show that activation of NF-κB is mutually exclusive with p53. And further that p53 activation correlates with NRF2 and JUN activation (Supplemental Figures 3I-K).

Third, we show that both the PRDX inhibitor Conoidin A, and the SRXN1 inhibitor J14, lower the dose of H₂O₂ required to switch from p53 to FOXO1 activation in MCF10A cells (Supplemental Figures 4E-G).

Other points:

Phosphorylated γH2AX analysis as proof of DNA damage: very superficial analysis. What type of DNA damage? Is the γH2AX bound to damaged DNA? Importantly, H₂O₂ has been shown to induce γH2AX phosphorylation independent of DNA damage.

This is a fair point, and we thank the reviewer for pointing out that γH2AX phosphorylation can occur independent of DNA damage. To verify that DNA damage is occurring in doses without p53 activation we took several approaches.

First, we measured and observed a dose-dependent increase in γH2AX foci in response to H₂O₂, suggesting γH2AX is bound to damaged DNA (Supplemental Figure S1D).

Second, we performed an alkaline and neutral comet assay (Figure 1F, G), and observed a dose-dependent increase in the percent of DNA in the tail for both assays. This supports an increase in DNA single, and double strand breaks. Increased DNA

damage occurred at high H₂O₂ concentrations where little to no p53 activation was observed by immunofluorescence.

Finally, we show that H₂O₂ treatment is dominant to treatment with the DNA damaging agent Neocarzinostatin (Figure 1H), supporting our model that p53 accumulation is blocked at high levels of H₂O₂.

That also speaks to the point above: What are the p53 targets under these conditions? Targets of FOXO1 and p53 need to be examined in an H₂O₂-time and dose-dependent manner.

As mentioned above, we feel that identifying the specific genes of each TF is beyond the scope of this study. We do provide bulk RNA-seq data that shows that typical FOXO1 and p53 target genes are upregulated in agreement with previous studies.

Regarding the PRDX1 FOXO3a interaction and FOXO1 regulation by PRDX1: PRDX1 was identified to interact via inter disulfides with FOXO3a C31 and C150. FOXO3a is the only FOXO member with a C150 residue. Experiments addressing the regulation of FOXO1 by PRDX1 are not shown.

We have not tested the role of specific cysteine residues in FOXO1 in this study. As mentioned below, in our previous study we found that nuclear FOXO1 was correlated with inhibition of pAKT-473, and thus unlike FOXO3a, nuclear accumulation of FOXO1 is likely due to AKT inhibition. In the discussion we cite a previous study that showed AKT is inactivated by H₂O₂ through a disulfide bond mechanism. Starting at Line 370: "In addition, AKT, which phosphorylates FOXO proteins causing cytoplasmic sequestration, is inhibited by disulfide bond formation between two cysteines in the protein, which leads to its inactivation and nuclear accumulation of FOXO." Though we have not directly tested that specific mechanism for FOXO1 shuttling.

PRDX biology: The authors show a PRDXSO₃ immunoblot. There are six mammalian PRDXs, four of which are 2-cysteine PRDXs, and they should interact with this antibody. Which isoform of PRDX shown in Figure 4 is being examined needs to be clarified.

Also, PRDX redox regulation by H₂O₂ includes dimer formation. Mono-sulfinic/sulfonic oxidized PRDX proteins may exist as dimers. Careful titration curves need to be shown documenting monomeric, dimeric, and overoxidized PRDX on non-reducing gels.

The reviewer is correct that the antibody we used in the first submission to measure PRDXSO₃ interacts with PRDX1/2/3/4. We apologize for not including this in the first submission and now mention this in the Figure legend for Figure 5B. As the reviewer suggested, we also performed titration curves using non-reducing western blots to

observe how H₂O₂ dose corresponds to the formation of dimers and hyperoxidized monomers for both PRDX1 and PRDX2 (Figure 5C, D). We observed dose-dependent monomer formation at doses where cells switch from p53 to FOXO1 activation in our immunofluorescence data. Together these data are consistent with hyperoxidation and inactivation of PRDX1 and PRDX2 as a key step in switching from p53 to FOXO1 activation.

This goes back to a point just mentioned above. Sulfiredoxin has other clients besides PRDX1. It also nitrosylated PRDX2. Which PRDX is responsible for FOXO1 translocation/activation?

Excellent points. To address the role of PRDX2 we used a doxycycline inducible shRNA to PRDX2 as we were unable to knockout PRDX2 (Supplemental figures 4J-N). Like our PRDX1 knockout cells, we observed an increase in FOXO1 active cells and a decrease in p53 active cells at low H₂O₂ concentrations when PRDX2 was knocked down. The effect was not as strong as in PRDX1 knockout cells, though we were only able to get a 2-fold knockdown of PRDX2. Thus, our current model is that PRDX1 and PRDX2 must be inactivated by H₂O₂ dependent hyperoxidation for FOXO1 to be activated. Though we do not know the direct mechanism for FOXO1 activation, we previously showed that FOXO1 activation is accompanied by a decrease phosphorylated Akt-473, suggesting Akt inactivation is a key step in activation of FOXO1 (see Figure 5: Lasick, Kathleen A., et al. "FOXO nuclear shuttling dynamics are stimulus-dependent and correspond with cell fate." *Molecular Biology of the Cell* 34.3 (2023),). How this occurs, and how PRDX1 and PRDX2 block this is unknown. We also agree that sulfiredoxin has other clients besides PRDX1 and PRDX2 and we mention this caveat in the discussion in lines 360-362: "One caveat of these data is that other proteins outside the PRDX family are hyperoxidized by H₂O₂, and are repaired by SRXN1, and we cannot rule out that these PRDX independent hyperoxidation events are involved in activation of p53 and FOXO1⁵⁵."

The H₂O₂ doses used are very high, unphysiologically high, for cell lines shown in the supplemental data.

New experiments that we performed for the resubmission and further literature review clarified some of the differences between low vs high H₂O₂ in MCF7 cells and the other cell lines tested. First, we engineered MCF10A to harbor FOXO1-mVenus and p53-mCherry reporters in order to observe the two phases of the H₂O₂ response (Supplemental Figure 2). This revealed that MCF10A cells show the two distinct phases at doses closer to what we observed in MCF7 cells (80 and 100 μM). One key difference was that the FOXO1 phase was much shorter in MCF10A cells. In our

original submission we measured FOXO1 and p53 activation in MCF10A cells 5 hours after H₂O₂ treatment. Based on our time-lapse data, this suggested we would miss most FOXO1 activation. Furthermore, MCF10A cells were more sensitive to H₂O₂, with 55% cell death at 100 μM vs, 33% for MCF7. Thus, we repeated the MCF10A immunofluorescence and measured FOXO1 and p53 activation 2 hours after H₂O₂ treatment and observed FOXO1 activation at lower doses than in the first submission (Supplemental Figure 1E).

As for A549 and U2OS cell lines, we repeated the immunofluorescence on these cell lines at 2 hours (vs 5) and did not observe a substantial difference between the 5 hour and 2 hour treatments, and decided not to include these data. A literature review revealed that A549 harbor KEAP1 mutations that lead to constitutive activation of NRF2 which is thought to result in resistance to oxidative stress. However no obvious mutations came up in our literature review for U2OS and thus we are unsure why these cells require much higher concentrations of H₂O₂ to activate FOXO1 and repress p53. We added the following to the Results section, starting at line 126 to highlight these discrepancies “However, the U2OS and A549 cell lines only activated FOXO1 at high levels of H₂O₂ (500 μM). A549 cells harbor mutations in KEAP1, which block its interaction with NRF2, leading to constitutive activation of NRF2 and is likely the reason this cell line is more resistant to H₂O₂³⁹. It is not clear why U2OS cells are more resistant to H₂O₂.”

Reviewer #2 (Remarks to the Author):

In this work, Jose and cols have explored the temporal coordination of transcription response to H₂O₂ stress. By using cutting-edge technologies, such as live cell microscopy, single cell ATAC and RNA sequencing, authors find a differential coordination along the time of the transcriptional program in response to different doses of H₂O₂. Authors propose that low concentrations of H₂O₂ mainly activate p53, whereas high concentrations induce 2 different phases: in a first phase, FOXO1, in the second phase, p53. Other transcription factors are activated in the first phase with FOXO1 or in the second phase with p53, but not with both. Finally, authors provide evidence about the role of peroxiredoxins in the control of this transcriptional program.

Work is of great interest to better understand the cell response to H₂O₂ but it shows different weaknesses that must be considered before publication of the manuscript.

We thank reviewer #2 for their interest in our work, for their careful reading of the manuscript and for their experimental suggestions.

MAJOR POINTS

1. One of the main concerns with this work is that the use of different concentrations of acute treatment with H₂O₂ also incorporates a variable, which is the time in which the H₂O₂ is acting, since H₂O₂ will disappear, either spontaneously, or by the cellular catalase. Indeed, the time for action of 40-50 μM H₂O₂ may be much shorter than the time in which the 200 μM H₂O₂ is acting. Did the authors analyze the decrease in the H₂O₂ concentrations in the culture medium along the time? The main results obtained at the beginning of the work, particularly the differential timing and intensity in the regulation of FOXO1 and p53, must be repeated by using a continuous system of H₂O₂ production, such as the combination of xanthine-xanthine oxidase (or other enzymatic continuous production of H₂O₂), where controlling the xanthine oxidase concentration the level of H₂O₂ will change, and this level will be maintained along the time (with enough xanthine as substrate). If results would not be identical, the focus of the work would be the temporal coordination of the transcription factor response to acute H₂O₂ stress. This experiment is even more relevant when looking at the results with ter-butyl-hydroperoxide, a mimetic of the H₂O₂ that is maintained in a constant concentration in the culture due to the inability of the catalase to metabolize it. Results with this compound are significantly different.

We thank the reviewer for this important point as the experiments we did to address their concerns had a significant impact on the manuscript and greatly improved the study. We break down our response to address each point.

First the reviewer asks:

“Did the authors analyze the decrease in the H₂O₂ concentrations in the culture medium along the time?”

We have not measured the decrease in H₂O₂ concentration in the culture medium in this study. In a previous study we measured H₂O₂ in MCF7 cells using the Hyper3 reporter (Figure 3: Lasick, Kathleen A., et al. "FOXO nuclear shuttling dynamics are stimulus-dependent and correspond with cell fate." *Molecular Biology of the Cell* 34.3 (2023)). In that study we found that bolus H₂O₂ treatment was cleared rapidly from cells, or within 1-1.5 hours, more or less, independent of dose. This is in line with other studies (Bilan DS, Pase L, Joosen L, Gorokhovatsky AY, Ermakova YG, Gadella TWJ, Grabher C, Schultz C, Lukyanov S, Belousov VV (2013). HyPer-3: a genetically encoded H₂O₂ probe with improved performance for ratiometric and fluorescence lifetime imaging. *ACS Chem Biol* 8, 535–542.) Also in our previous study, adding the antioxidant N-Acetyl-Cysteine (NAC) 1 hour after H₂O₂ treatment, blocked FOXO1

activation, but adding it 2 hours after H₂O₂ had no effect. Therefore, for the bolus H₂O₂ treatments, activation of each group of transcription factors occurs well after H₂O₂ is cleared from cells as our immunofluorescence assays are performed 5 hours after H₂O₂ treatment.

Next the reviewer asks for experiments using continuous H₂O₂ production.

“The main results obtained at the beginning of the work, particularly the differential timing and intensity in the regulation of FOXO1 and p53, must be repeated by using a continuous system of H₂O₂ production, such as the combination of xanthine-xanthine oxidase (or other enzymatic continuous production of H₂O₂), where controlling the xanthine oxidase concentration the level of H₂O₂ will change, and this level will be maintained along the time (with enough xanthine as substrate). If results would not be identical, the focus of the work would be the temporal coordination of the transcription factor response to acute H₂O₂ stress”

Again, this was an excellent suggestion. We used Glucose Oxidase for continuous H₂O₂ production (see figure 3). Performing a dose-response with glucose oxidase showed there is still temporal ordering in the activation of p53 and FOXO1, however under continuous H₂O₂ production the order of activation was reversed; p53 was activated first, and then after a delay FOXO1 was activated. Interestingly, When FOXO1 was activated under continuous H₂O₂ production, p53 levels neither increased nor decreased but remained flat and elevated from baseline. The description of this is provided in the results section titled “p53 accumulation precedes FOXO1 activation in response to continuous H₂O₂ production” lines 172-204. In the discussion, lines 381-388 we speculate on why the order of activation for p53 and FOXO1 differs between acute and continuous H₂O₂ production. And finally, we updated the cartoon in Figure 7 to reflect these differences. We are working on a follow-up study to better understand the difference between these two treatments and thank the reviewer for this suggestion.

Finally the reviewer mentioned:

“This experiment is even more relevant when looking at the results with ter-butyl-hydroperoxide, a mimetic of the H₂O₂ that is maintained in a constant concentration in the culture due to the inability of the catalase to metabolize it. Results with this compound are significantly different.”

We are not sure why Tert-butyl-hydroperoxide behaved differently in our experiments than either bolus H₂O₂ or continuous H₂O₂. Though speculative, this might be due to tBHP increased capacity for lipid peroxidation as we only saw modest increases in lipid peroxidation and only at H₂O₂ doses > 150 μM by 4-HNE staining (Supplemental Figure 1K).

2. Most of the work (with the exception of a punctual experiment at the beginning of the work) has been made in only one cell line, which is a tumoral cell. Do authors consider that other cells, particularly non tumoral cells, will respond in the same way? In order to better evaluate the relevance of the work, at least some specific important results presented (such as the additional H₂O₂ induced transcription factors activated with either FOXO1 or p53 or the role of the peroxiredoxin/sulfiredoxin system) must be demonstrated in a non-tumoral cell, such as the MCF10A used in Fig S1.

We have added more supporting data using MCF10A cells as the authors suggested. First, we created an MCF10A FOXO1-mVenus, p53-mCherry cell line and performed time lapse imaging under acute H₂O₂ stress. We observed a dose-dependent increase in the duration of nuclear FOXO1 and delay of p53 activation until FOXO1 exits the nucleus (Supplemental Figures 2I-Q).

Second, using immunofluorescence in MCF10A cells, we show that activation of NF-κB is mutually exclusive with p53. And further that p53 activation correlates with NRF2 and JUN activation (Supplemental Figures 3I-K).

Third, we show that both the PRDX inhibitor Conoidin A, and the SRXN1 inhibitor J14, lower the dose of H₂O₂ required to switch from p53 to FOXO1 activation in MCF10A cells (Supplemental Figures 4E-G).

3. The strong increase in nuclear p53 presented in Fig. 1A (40 uM H₂O₂) cannot be observed in the video Supplemental Movie 1. Stopping the movie at 5 h the image is more similar to that presented for untreated cells. There is no clear correlation, neither, in the 80 uM or 100 uM, when stopping the movie at 5 h. The only result that clearly correlates is the 200 uM concentration.

We see what you are pointing out here. There are a couple reasons for the difference in p53 activation between the experiments in Figure 1A, the Supplemental Movies, and the data from these movies in Figure 2.

First, we used 96-well plates for the data in Figure 1 and all other immunofluorescence experiments, and used 12 well glass bottom plates for the time-lapse imaging experiments as these plates are easier to add H₂O₂ and other drugs while in our microscope. We note that the percentage of FOXO1 active and p53 active cells for a given H₂O₂ dose is sensitive to cell density (See Supplemental Figure 1B for quantification of this). Although we tried to match cell densities between the two types of experiments the inevitable differences and the bimodality of the response

means that there is not a 1:1 mapping of the H₂O₂ concentrations from the immunofluorescence data and the live-cell imaging data. For supplemental movie 1 (50 μM) dose, since there is almost no FOXO1 response, p53 is activated immediately in a series of oscillations. The first peak in p53 levels is ~3 hours in (Figure 2B), followed by a trough at 5 hours (which the reviewer noted), then peaks again at ~8-9 hours. This is identical to what has been shown for p53 dynamics in response to DNA double strand breaks (Lahav, G., Rosenfeld, N., Sigal, A. *et al.* Dynamics of the p53-Mdm2 feedback loop in individual cells. *Nat Genet* **36**, 147–150 (2004)).

Second, even given the differences between the experiments in Figure 1 and 2, we agree with the reviewer that the strong increase in p53 in Figure 1 does not appear as strong in our time-lapse images. We believe this is due to the limitations of live-cell imaging. Given that live-cell imaging must occur in culture media, there is higher background signal. In addition, fluorescent proteins like mCherry which we used to tag p53, have poor spectral properties such as brightness and quantum yield, especially when compared to alexa-fluor dyes used for immunofluorescence. Thus, a decrease in fluorescent signal coupled with an increase in background is likely the difference in the magnitude of the response.

4. The only functional analysis of oxidation of intracellular components in this work is γH2AX, which only refers to DNA damage and which was monitored for comparison with the situation of p53 activation. Do different doses of H₂O₂, or different times, affect differentially to oxidation of other components, such as proteins or lipids?

Excellent point, to address this we measured protein oxidation using maleimide-488 and compared this to total protein levels using Alexa Fluor 647 carboxylic acid succinimidyl ester (Supplemental Figure S1J). We see substantial protein oxidation at all doses tested. We also used 4-HNE staining for lipid peroxidation (Supplemental Figure S1K). Interestingly we only see an increase in lipid peroxidation at doses of H₂O₂ 150 μM and above (Erastin was used as a positive control).

MINOR POINTS

1. From the beginning, authors focus on FOXO1 and p53 because both are activated by H₂O₂ regulating cell-cycle arrest and apoptosis. References for this assessment must be incorporated in the Introduction section. And not only FOXO1 or p53 are activated by H₂O₂ to regulate proliferation and cell death. Authors must better justify why to initiate the work focusing on these transcription factors.

We have now added citations on both p53 and FOXO's role in H₂O₂ induced cell-cycle arrest and apoptosis. In addition, we cite a review that discusses the overlap in target genes of FOXO and p53 as well as other similarities between the two transcription factors

2. Authors mention in the discussion section that: “.. pentose phosphate pathway and nucleotide biosynthesis in the second transcription factor phase.....Upregulation of these genes are likely important for completing DNA replication and DNA repair”. Pentose phosphate cycle will be relevant for NADPH production, additionally to ALDH3A1, ME1 that authors indicate.

Excellent point. For this version of the discussion, we decided to remove discussion of specific target genes but kept discussion of the broad categories upregulated.

Reviewer #3 (Remarks to the Author):

The manuscript “Temporal coordination of the transcription factors response to H₂O₂ stress” described how activation of different transcription factors is tightly and temporally coordinated in response to H₂O₂. The authors provided compelling evidence that p53 and FOXO1 are differentially activated in response to different levels of H₂O₂. Low levels of H₂O₂ activate p53 but not FOXO1, while high levels of H₂O₂ lead to two phases of transcription factor activation. FOXO1 is activated in the first phase, followed by p53 activation in the second phase in a H₂O₂ dose dependent manner. They further showed that other transcription factors (RelA, NFAT1, NRF2 and JUN) are also activated coordinately and differentially with FOXO1 and p53 in response to different levels of H₂O₂. The two temporal phases are associated with distinct gene expression changes corresponding to the changes in transcription factors. Lastly, they provided evidence that PRDX dependent redox relay dictates the activation of FOXO1 and p53, suggesting that peroxiredoxins control the type and timing of transcription factor activation. A similar PRDX-dependent redox relay mechanism exists in yeast, and now this study provided evidence for such mechanism in mammalian cells in response to H₂O₂. The temporal and coordinated activation of different transcription factors in response to different levels of H₂O₂ leads to gene expression changes that are necessary for the cytoprotective processes, providing a mechanistic understanding. Overall, the studies were well designed, methodologies are sound and detailed, and the data were strong in support of their conclusions.

We thank reviewer #3 for their careful reading of our manuscript, their complements about the design and analysis of the data and their helpful suggestions.

There are a few concerns that need to be addressed.

1. What defines low vs. high H₂O₂ dose? Low dose used in MCF10A, A549, U2OS cells (200-300 μM) is the high dose in MCF7. What causes the discrepancy? Should the intracellular H₂O₂ level be measured to define low vs. high dose? The doses used were quite high, causing significant cell death (~98% with 300 μM), which may complicate the interpretation.

New experiments that we performed for the resubmission and further literature review clarified some of the differences between low vs high H₂O₂ in MCF7 cells and the other cell lines tested. First, we engineered MCF10A to harbor FOXO1-mVenus and p53-mCherry reporters in order to observe the two phases of the H₂O₂ response (Supplemental Figure 2). This revealed that MCF10A cells show the two distinct phases at doses closer to what we observed in MCF7 cells (80 and 100 μM). One key difference was that the FOXO1 phase was much shorter in MCF10A cells. In our original submission we measured FOXO1 and p53 activation in MCF10A cells 5 hours after H₂O₂ treatment. Based on our time-lapse data, this suggested we would miss most FOXO1 activation. Furthermore, MCF10A cells were more sensitive to H₂O₂, with 55% cell death at 100 μM vs, 33% for MCF7. Thus, we repeated the MCF10A immunofluorescence and measured FOXO1 and p53 activation 2 hours after H₂O₂ treatment and observed FOXO1 activation at lower doses than in the first submission (Supplemental Figure 1E).

As for A549 and U2OS we repeated the immunofluorescence on these cell lines at 2 hours (vs 5) and did not observe a substantial difference between the 5 hour and 2 hour treatments and decided not to include these data. A literature review revealed that A549 harbor KEAP1 mutations that lead to constitutive activation of NRF2 which is thought to result in resistance to oxidative stress. However no obvious mutations came up in our literature review for U2OS and thus we are unsure why these cells require much higher concentrations of H₂O₂ to activate FOXO1 and repress p53. We added the following to the Results section, starting at line 126 to highlight these discrepancies “However, the U2OS and A549 cell lines only activated FOXO1 at high levels of H₂O₂ (500 μM). A549 cells harbor mutations in KEAP1, which block its interaction with NRF2, leading to constitutive activation of NRF2 and is likely the reason this cell line is more resistant to H₂O₂³⁹. It is not clear why U2OS cells are more resistant to H₂O₂.”

2. Is activated p53, JUN, or NRF2 in the same cells? Same question for FOXO1, RelA and NFAT1. ATAC-seq cannot distinguish each individual cell. From Figure 2, p53 is activated in

~70% of cells at 40 μ M, ~60% at 60 μ M, but nuclear Jun is in ~25% at 40 μ M and nuclear NRF2 is in ~20% of cells at 60 μ M, suggesting that these transcription factors may be activated in different cells. Measuring nuclear p53 and JUN/NRF2 simultaneously will answer this question, and same is true for the FOXO1/NFAT1 group. Answers to these questions will strengthen the transcriptomic data for the coordinated gene expression changes.

Great question/suggestion. We repeated the IF for each of the TFs to measure FOXO1 activation with RelA and NFAT, and p53 activation with NRF2 and JUN (Supplemental Figure 3E-H). For the most part there is a lot of overlap in activation with some caveats. For example, most RelA active cells have active FOXO1, yet there are many FOXO1 active cells with no RelA. This might be due to the fact that RelA is known to oscillate in and out of the nucleus under some stimuli while FOXO1 remains sustained in the nucleus during the FOXO1 phase. Or it could just be that only a subset of cells activate RelA. NRF2 and p53 had a similar relationship. Following up on these differences using RelA/Nrf2 reporters would be interesting and something we are planning to do in a follow-up study.

3. Figure 2 and Figure S2: autocorrelation assay is used but lacks the explanation how it is done in Materials and Methods.

Thanks for catching this! We have added details on how we performed the autocorrelation analysis in the materials and methods.

REVIEWER COMMENTS

Reviewer #1 (Remarks to the Author):

The authors went to great lengths to address my concerns.

One important issue however that is still not being addressed is the confirmation of their findings in an intrinsic H₂O₂-producing system. This provides the necessary biological relevance; without it, this study is simply limited, especially for a Nature Communications article.

DAAO expression constructs are available as adenoviral vectors (Addgene) which should make expression rather easy.

At least, a cytoplasmic OR mitochondrial expression vector should be tested to add the necessary physiological relevance.

Reviewer #2 (Remarks to the Author):

Authors have correctly addressed all my concerns. In the revised version, with huge new experimental work incorporated, conclusions are strongly supported by the results.

Reviewer #3 (Remarks to the Author):

The authors have addressed my comments adequately and I recommend the manuscript for publication.

REVIEWER COMMENTS

Response to reviewers in bold

Reviewer #1 (Remarks to the Author):

The authors went to great lengths to address my concerns.

One important issue however that is still not being addressed is the confirmation of their findings in an intrinsic H₂O₂-producing system. This provides the necessary biological relevance; without it, this study is simply limited, especially for a Nature Communications article.

DAAO expression constructs are available as adenoviral vectors (Addgene) which should make expression rather easy.

At least, a cytoplasmic OR mitochondrial expression vector should be tested to add the necessary physiological relevance.

We thank reviewer #1 for their helpful comments and suggestions and are glad that we addressed many of their concerns. While we agree that DAAO experiments would be interesting, we believe that these experiments would delay publication and increase the costs without adding significant biological relevance or increasing the scientific impact of the publication. The stated reason for the DAAO experiments is to provide confirmation of our findings using an intrinsic H₂O₂ producing system. However, we believe the current manuscript already shows intrinsic ROS leads to the key findings. We have shown that menadione exposure results in mutually exclusive activation of p53 and FOXO1 in a dose-dependent manner similar to H₂O₂ and Glucose Oxidase treatment (Supplemental Figure S1K in the current version of the manuscript). Menadione generates intracellular ROS through redox cycling (Criddle et al). Moreover, menadione is known to create intracellular H₂O₂ as the toxic effects of menadione are reversed by overexpression of catalase targeted to the cytoplasm or the mitochondria (Loor et al).

We have added the following language in the manuscript to point this out and have cited the relevant studies.

On page 4 lines 131-133 we write:

"Menadione induces ROS through redox cycling which creates superoxide radicals that dismutate to H₂O₂, and menadione induced cell death can be suppressed by overexpressing catalase^{41,42}. Thus, it is possible that mutually exclusive activation of p53 and FOXO1 is specific to H₂O₂."

References:

Criddle DN, Gillies S, Baumgartner-Wilson HK, Jaffar M, Chinje EC, Passmore S, Chvanov M, Barrow S, Gerasimenko OV, Tepikin AV, Sutton R, Petersen OH. Menadione-induced reactive oxygen species generation via redox cycling promotes apoptosis of murine pancreatic acinar cells. *J Biol Chem*. 2006 Dec 29;281(52):40485-92. doi: 10.1074/jbc.M607704200. Epub 2006 Nov 6. PMID: 17088248.

Loor G, Kondapalli J, Schriewer JM, Chandel NS, Vanden Hoek TL, Schumacker PT. Menadione triggers cell death through ROS-dependent mechanisms involving PARP activation without requiring apoptosis. *Free Radic Biol Med*. 2010 Dec 15;49(12):1925-36. doi: 10.1016/j.freeradbiomed.2010.09.021. Epub 2010 Oct 27. PMID: 20937380; PMCID: PMC3005834.

Reviewer #2 (Remarks to the Author):

Authors have correctly addressed all my concerns. In the revised version, with huge new experimental work incorporated, conclusions are strongly supported by the results.

We are pleased that we have addressed all of your concerns and thank reviewer #2 for their helpful comments and suggestions.

Reviewer #3 (Remarks to the Author):

The authors have addressed my comments adequately and I recommend the manuscript for publication.

We also thank reviewer #3 for the thoughtful comments and suggestions and are glad that we adequately addressed them.

REVIEWERS' COMMENTS

Reviewer #1 (Remarks to the Author):

The text in the paper needs to discuss that while it is true that catalase can scavenge Menadione derived ROS, Menadione treatment does not replace an intracellular ROS producing system, which is a potential pitfall.

REVIEWER COMMENTS

Response to reviewers in bold

REVIEWERS' COMMENTS

Reviewer #1 (Remarks to the Author):

The text in the paper needs to discuss that while it is true that catalase can scavenge Menadione derived ROS, Menadione treatment does not replace an intracellular ROS producing system, which is a potential pitfall.

We added the following paragraph to the discussion on page 10 lines 398-406:

“One pitfall of our study is that we have not explored TF activation in response to a purely intracellular source of ROS. Menadione exposure results in mutually exclusive activation of FOXO1 and p53 similar to H₂O₂ (Figure S1K). This suggests that H₂O₂ produced within the cell elicits a similar TF response as extracellular H₂O₂ exposure as menadione induces ROS through redox cycling, and menadione induced cell death is suppressed by intracellular catalase expression^{41,42}. However, we cannot exclude the potential for menadione producing ROS through reactions with the media. Future studies using d-amino acid oxidase enzymes localized to specific cellular compartments will be useful for determining whether intracellular H₂O₂ behaves similar to extracellular H₂O₂ exposure, and if the TF response is dependent on the location of H₂O₂ production within the cell.”